# Urban areas as hotspots for bees and pollination but not a panacea for all insects

Panagiotis Theodorou [1,2,3]*, Rita Radzevičiūtė[1,4,5,6], Guillaume Lentendu [7,8], Belinda Kahnt[1,2], Martin Husemann[1,9], Christoph Bleidorn[2,10], Josef Settele [2,3,11], Oliver Schweiger [3], Ivo Grosse[2,12], Tesfaye Wubet [2,8], Tomás E. Murray[1,13] & Robert J. Paxton [1,2]

Urbanisation is an important global driver of biodiversity change, negatively impacting some species groups whilst providing opportunities for others. Yet its impact on ecosystem services is poorly investigated. Here, using a replicated experimental design, we test how Central European cities impact flying insects and the ecosystem service of pollination. City sites have lower insect species richness, particularly of Diptera and Lepidoptera, than neighbouring rural sites. In contrast, Hymenoptera, especially bees, show higher species richness and flower visitation rates in cities, where our experimentally derived measure of pollination is correspondingly higher. As well as revealing facets of biodiversity (e.g. phylogenetic diversity) that correlate well with pollination, we also find that ecotones in insect-friendly green cover surrounding both urban and rural sites boost pollination. Appropriately managed cities could enhance the conservation of Hymenoptera and thereby act as hotspots for pollination services that bees provide to wild flowers and crops grown in urban settings.

[1] General Zoology, Institute for Biology, Martin Luther University Halle-Wittenberg, Hoher Weg 8, 06120 Halle, Germany. [2] German Centre for Integrative Biodiversity Research (iDiv) Halle-Jena-Leipzig, Deutscher Platz 5e, 04103 Leipzig, Germany. [3] Department of Community Ecology, Helmholtz Centre for Environmental Research-UFZ, Theodor-Lieser-Strasse 4, 06120 Halle, Germany. [4] Molecular Evolution and Animal Systematics, Institute of Biology, University of Leipzig, Talstrasse 33, 04103 Leipzig, Germany. [5] ESCALATE, Department of Computational Landscape Ecology, Helmholtz Centre for Environmental Research – UFZ, Permoserstrasse 15, 04318 Leipzig, Germany. [6] Life Sciences Center, Vilnius University, Saulėtekio al. 7, 10223 Vilnius, Lithuania. [7] Department of Ecology, University of Kaiserslautern, Erwin-Schroedinger Street Building 14, 67663 Kaiserslautern, Germany. [8] Department of Soil Ecology, Helmholtz Centre for Environmental Research-UFZ, Theodor-Lieser-Strasse 4, 06120 Halle, Germany. [9] Centrum für Naturkunde, University of Hamburg, Martin-Luther-King-Platz 3, 20146 Hamburg, Germany. [10] Animal Evolution and Biodiversity, Johann-Friedrich-Bluemenbach Institute for Zoology and Anthropology, Georg-August-University Göttingen, Untere Karspüle 2, 37073 Göttingen, Germany. [11] Institute of Biological Sciences, University of the Philippines Los Baños, College, Laguna 4031, Philippines. [12] Institute of Computer Science, Martin Luther University Halle-Wittenberg, Von-Seckendorff-Platz 1, 06120 Halle, Germany. [13] National Biodiversity Data Centre, Beechfield House, WIT West Campus, X91 PE03 Waterford, Ireland. *email: panatheod@gmail.com

Insects are a vital component of terrestrial biodiversity, underpinning important ecosystem services such as pollination, soil formation and control of herbivorous pest species[1]. Yet they are also of considerable conservation concern, highlighted by a >75% decline in flying insect biomass over the past 27 years in German nature reserves[2], a 78% decline in arthropod abundance and 34% decline in arthropod species richness between 2008 and 2017 in German grasslands[3] and by a 33% range decline of bee and hover fly species between 1980 and 2013 in Britain[4]. Anthropogenic land use change is likely the main driver of terrestrial biodiversity decline[5], including that of insects[6]. One such change, urbanisation, has been identified as a threat to global biodiversity[7], including pollinator biodiversity[8], and ecosystem services[9]. In an increasingly human-dominated world of expanding cities[7], integrating the conservation of biodiversity and ecosystem services into urban planning and practices is therefore of importance for a sustainable future[10]. In juxtaposition to the city, the rural landscape is often dominated by agricultural land use (e.g. total German land cover: 51.6% agricultural versus 13.7% urban/transport[11]), which is associated with markedly reduced insect pollinator diversity[2,3].

Pollination is a crucial ecosystem service not only in natural but also in agricultural and urban ecosystems. An estimated 87.5% of angiosperm species are dependent on animal vectors for pollination and seed set[12] whilst the current economic value of pollination to world agriculture is ca. US$ $235–557 \times 10^9$ per annum at 2015 prices[13]. Diverse land uses within European cities can be very rich in native flowering plant species[14,15] and there is also an increasing interest in the potential of (outdoor) urban agriculture in ensuring food security[16]. Yet the impact of urbanisation on the pollination of wild and cultivated plants remains poorly known[17]. We also lack direct comparison with rural sites, which are typically dominated by agricultural land use and where pollinators are vital for crop pollination, yet where agricultural intensification is thought to result in reduced provision of a range of ecosystem services provided by insects, including pollination[18].

Urbanisation has been shown to be associated with a change in pollinator community composition[19,20], including a decrease in insect pollinator species richness[19] and abundance[20]. Other studies have, in contrast, shown urban areas to have neutral or even positive effects on biodiversity, including some insect pollinator groups, and particularly wild bee species[21,22]. Botanical gardens, allotments and residential gardens and urban vacant lots[15,23] may be particularly rich in wild bee species. In rural areas of Europe, in contrast, agricultural land use is associated with reduced insect biodiversity, including compromised growth and reproductive success of pollinators (e.g. *Bombus terrestris*[24]), and reduced wild plant pollination[22].

The extent to which pollinator biodiversity is impacted by urbanisation, as for other anthropogenic impacts such as agricultural land use, likely depends on the intensity of land use, the spatial scale of investigation, and the taxonomic group studied[25]. Habitat change and moderate disturbance within either urban or rural environments could potentially increase landscape heterogeneity and the availability of suitable pollinator habitats and resources, thereby increasing niche diversity and enhancing insect pollinator diversity[26,27]. Two aspects of landscape heterogeneity, namely composition (e.g. total proportion of each habitat type) and configuration of habitats (e.g. habitat fragmentation), are expected to have distinct effects on different pollinator groups or ecosystem processes[28]. Additionally, insect pollinator communities respond positively to small-scale (patch) habitat features associated with nesting and food (floral) resources[22,29], often irrespective of land use change[22,30]. Thus, land use change and disturbance in moderately urbanised and rural areas, with increased cover and connectivity of semi-natural vegetation providing abundant foraging and nesting resources, could potentially support pollinator biodiversity[26,28,31], possibly explaining why bee biodiversity is higher in some—though not all—urban versus rural sites[21,22]. Yet although we have an increasing understanding of how urbanisation and rural land use change can impact certain pollinator taxa, it remains unclear how these changes translate into altered pollination.

Most studies addressing the impact of urbanisation or agricultural land use on pollinator biodiversity have so far focused on species richness and disregarded the variability between species, including their different evolutionary histories and ecological functions. Phylogenetic diversity provides a way to capture these species differences through the assessment of species' evolutionary relationships and can be used as a proxy for phenotypic diversity and evolutionary potential of a community[32,33]. Phylogenetic diversity can also be related to ecosystem functioning and ecosystem services[33] and consequently could act as an important metric in nature conservation[34]. In support of this view, bee phylogenetic diversity and the pollination service they provide to apple crops both drop with increasing agricultural land cover in the rural landscape[35]. Despite the recognised predictive conservation potential of phylogenetic diversity, we do not know how urbanisation affects the phylogenetic diversity of pollinator communities and, furthermore, how it affects the ecosystem service of pollination that they provide.

Here we used a paired study design (Fig. 1) at flower-rich sites in nine independent German cities and nine nearby, equivalent, flower-rich rural sites explicitly to test the impact of urbanisation on pollinator biodiversity and the ecosystem service of pollination. We used metabarcoding to evaluate flying insect diversity beyond just the bees. To quantify pollination, we used 'pollinometers': potted, greenhouse-raised, insect-pollinator dependent red clover (*Trifolium pratense*) plants, which allowed us to derive an objective measure of the ecosystem service of pollination. Our results reveal that German cities harbour lower insect species richness, especially for Diptera and Lepidoptera, than rural areas. In contrast, we find higher Hymenoptera species richness and flower visitation rates by bees in urban areas, positively driven by edge density of green spaces, and associated with an increase in pollination. Moreover, we find that phylogenetic diversity can be a valuable predictor of the effects of biodiversity on ecosystem services, highlighting evolutionary history as a focus of attention to improve our understanding of the consequences of biodiversity loss.

## Results

**Insect community diversity in urban versus rural sites**. We assessed flying insect diversity by pan-trapping and metabarcoding at nine urban sites and nine adjacent rural sites (Fig. 1), each of which we considered optimal for flower-visiting insects. Our rural sites supported a higher overall Insecta species richness and biomass compared to urban sites (Fig. 2a, b), differences which remained significant when controlling for local (patch) and landscape variables (GLMM, $Z = 4.301$, $P < 0.001$, Fig. 2a; LMM, $t = 2.387$, $P = 0.048$, Fig. 2b). Phylogenetic species variability (PSV) and mean nearest taxon distance (MNTD) did not differ between rural and urban sites (GLMM, $Z = 0.00$, $P = 1.00$, Fig. 2c; LMM, $t = 0.319$, $P = 0.758$, Supplementary Fig. 1; respectively). But this pattern belies differences among insect orders. Specifically, rural sites supported higher Lepidoptera and Diptera operational taxonomic unit (OTU) richness (GLMMs, $Z = 7.106$, $P < 0.001$; $Z = 2.160$, $P = 0.031$; Fig. 2d, e; respectively), though richness did not differ between rural and urban sites for hover flies (Syrphidae: GLMM, $Z = 0.231$, $P = 0.822$; Fig. 2f) or for Coleoptera (GLMM, $Z = 0.931$, $P = 0.352$; Fig. 2g).

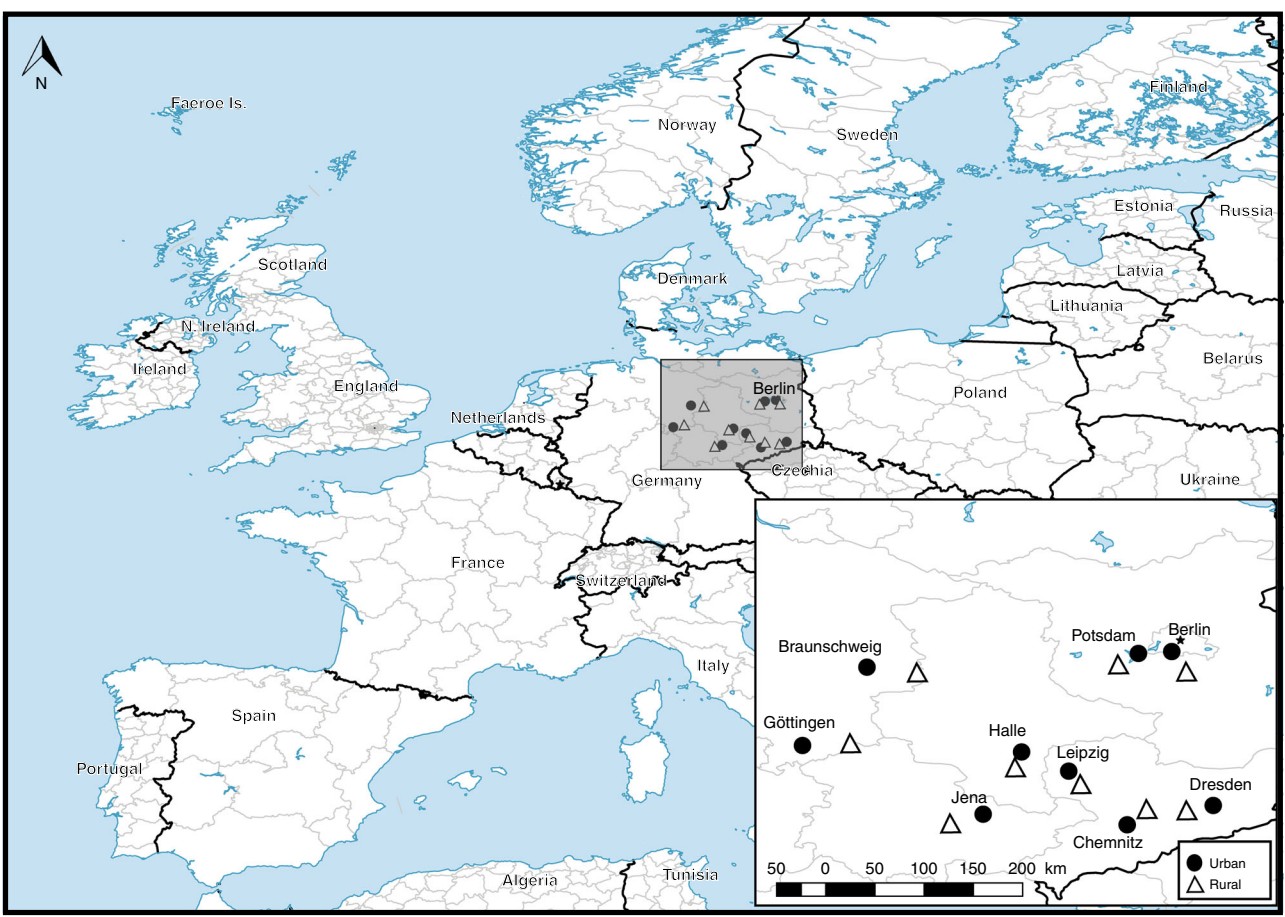

**Fig. 1 Location of the 18 study sites within Germany.** Cities are labelled; black circles correspond to urban sites and white triangles correspond to rural sites. This map was created with Natural Earth, free vector and raster map data @ naturalearthdata.com using QGIS Desktop. This map is licenced under Creative Commons Attribution-ShareAlike 3.0 licence (CC BY-SA) https://creativecommons.org/licenses/by-sa/3.0/.

However, compared to rural sites, urban sites supported a significantly higher Hymenoptera OTU richness (GLMM, $Z = 1.979$, $P = 0.047$; Fig. 2h), especially that of bees (Anthophila, a subset of Hymenoptera: GLMM, $Z = 2.737$, $P = 0.006$; Fig. 2i).

The honeypot effect might in part account for differences we detected between urban versus paired rural sites in insect pollinator community diversity. Yet local and landscape covariates included in our statistical models did not differ markedly between site type. Firstly, total flower abundance did not differ between urban versus rural sites (LMM, $t = 0.403$, $P = 0.697$; Supplementary Table 1). Even though species richness of flowering plants was higher at urban sites (GLMM, $Z = 3.350$, $P < 0.001$; Supplementary Table 1), our data suggest that urban and rural sites were similar in their capacity to attract flying insects from afar. Secondly, landscape factors that might be particularly associated with high insect community biodiversity, namely total green cover and edge density (habitat configurational heterogeneity; see below), did not differ between urban versus rural sites (LMM, $t = -0.080$, $P = 0.938$; LMM, $t = 0.487$, $P = 0.632$, respectively; Supplementary Fig. 2). These results suggest that, if a honeypot effect had impacted the insect communities that we measured, then it likely impacted both urban and rural sites equally.

Relationships between insect diversity and environmental (local and landscape) habitat features varied across insect orders, though insect diversity often increased with habitat heterogeneity and edge density in both urban and rural ecosystems. Urban sites with higher edge density supported higher Hymenoptera and bee (Anthophila) OTU richness (GLM; $Z = 2.544$, $P = 0.011$; $Z = 3.160$, $P = 0.001$; respectively; Fig. 3, Supplementary Table 2). Furthermore, urban sites with higher habitat diversity supported higher Coleoptera OTU richness (GLM; $Z = 3.225$, $P = 0.001$; Fig. 3, Supplementary Table 2). At urban sites, Diptera OTU richness increased not only with edge density but also with residential cover (GLM; $Z = 2.436$, $P = 0.014$; $Z = 2.172$, $P = 0.029$; respectively; Fig. 3, Supplementary Table 2). Due to low sample size (mean = 1.44 ± 1.01 SD), Lepidoptera OTUs richness was not modelled within urban sites. We did not find any significant predictors of hover flies (Syrphidae) OTU richness in urban ecosystems.

In rural areas, Hymenoptera and bee (Anthophila) OTU richness increased with higher edge density (GLM; $Z = 2.530$, $P = 0.011$; $Z = 2.477$, $P = 0.013$; respectively; Fig. 3, Supplementary Table 2) and decreased with the proportion of arable land (GLM; $Z = -3.001$, $P = 0.002$; $Z = -2.112$, $P = 0.034$; respectively; Fig. 3, Supplementary Table 2). Coleoptera OTU richness at rural sites increased with higher flower richness (GLM; $Z = 3.352$, $P < 0.001$; Fig. 3, Supplementary Table 2) and higher habitat diversity (GLM; $Z = 2.471$, $P = 0.013$; Fig. 3, Supplementary Table 2). Diptera and Lepidoptera OTU richness increased with flower richness (GLM; $Z = 2.517$, $P = 0.011$; $Z = 3.128$, $P = 0.001$; respectively; Fig. 3, Supplementary Table 2). We did not find any significant predictors of hover flies (Syrphidae) OTU richness in rural ecosystems.

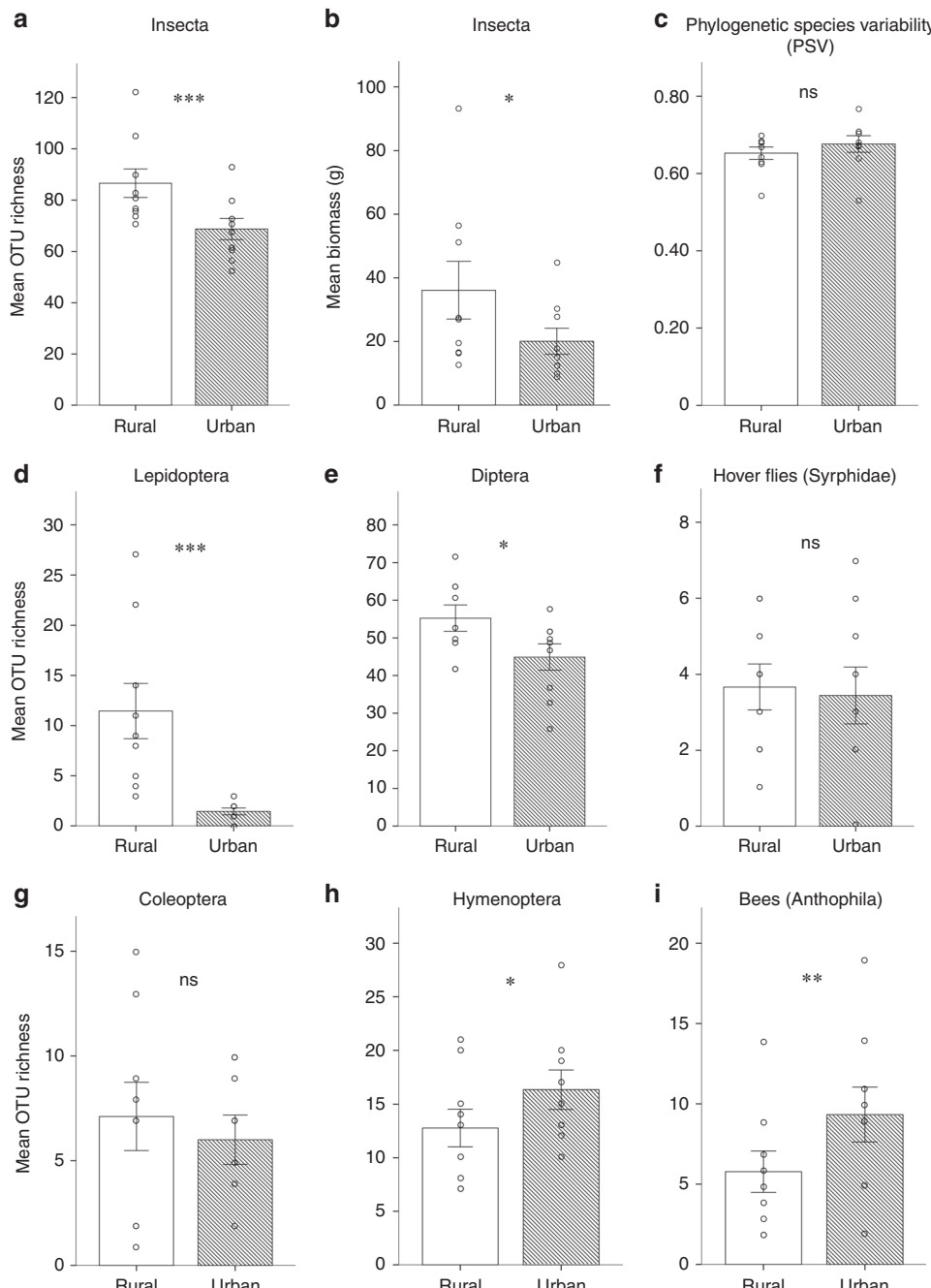

**Fig. 2 Insect biodiversity at flower-rich rural versus paired flower-rich urban sites.** Insect biodiversity at $N = 9$ flower-rich rural versus $N = 9$ paired flower-rich urban sites as **a** mean species richness (number of OTUs) of Insecta, **b** insect biomass, **c** insect phylogenetic species variability (PSV), as well as species richness of **d** Lepidoptera, **e** Diptera, **f** hover flies (Syrphidae) (a subset of (**e**)), **g** Coleoptera, **h** Hymenoptera, and **i** bees (Anthophila) (a subset of (**h**)); means ± SE are shown; GLMM and LMM: ns, not significant, one star, $P < 0.05$, two stars, $P < 0.01$; three stars, $P < 0.001$.

Both total insect composition and Hymenoptera community composition differed between rural and urban ecosystems (*adonis* all insects: $F = 1.574$, $R^2 = 0.089$, $P = 0.003$; Hymenoptera only: $F = 1.692$, $R^2 = 0.095$, $P = 0.004$, respectively; Supplementary Fig. 3); several species were found in both urban and rural sites, e.g., *Bombus terrestris* and *Lasioglossum calceatum*, but others were restricted to few sites, often within the same ecosystem, e.g., *Heriades truncorum* in urban sites, *Dasypoda hirtipes* in rural sites (see Supplementary Dataset). Overall insect communities and Hymenoptera communities were more homogeneous in urban compared to rural sites (LMM, all insects: $t = 2.587$, $P = 0.032$; only Hymenoptera: $t = 4.312$, $P = 0.002$; Supplementary Fig. 3).

**Pollination of *T. pratense* in urban versus rural sites.** We experimentally quantified provision of the ecosystem service of pollination using 'pollinometers', potted, glasshouse-raised plants of native red clover, *Trifolium pratense*. There was greater provision of the ecosystem service of pollination at flower-rich urban versus flower-rich rural sites; *T. pratense* plants produced more seeds in urban compared to rural sites

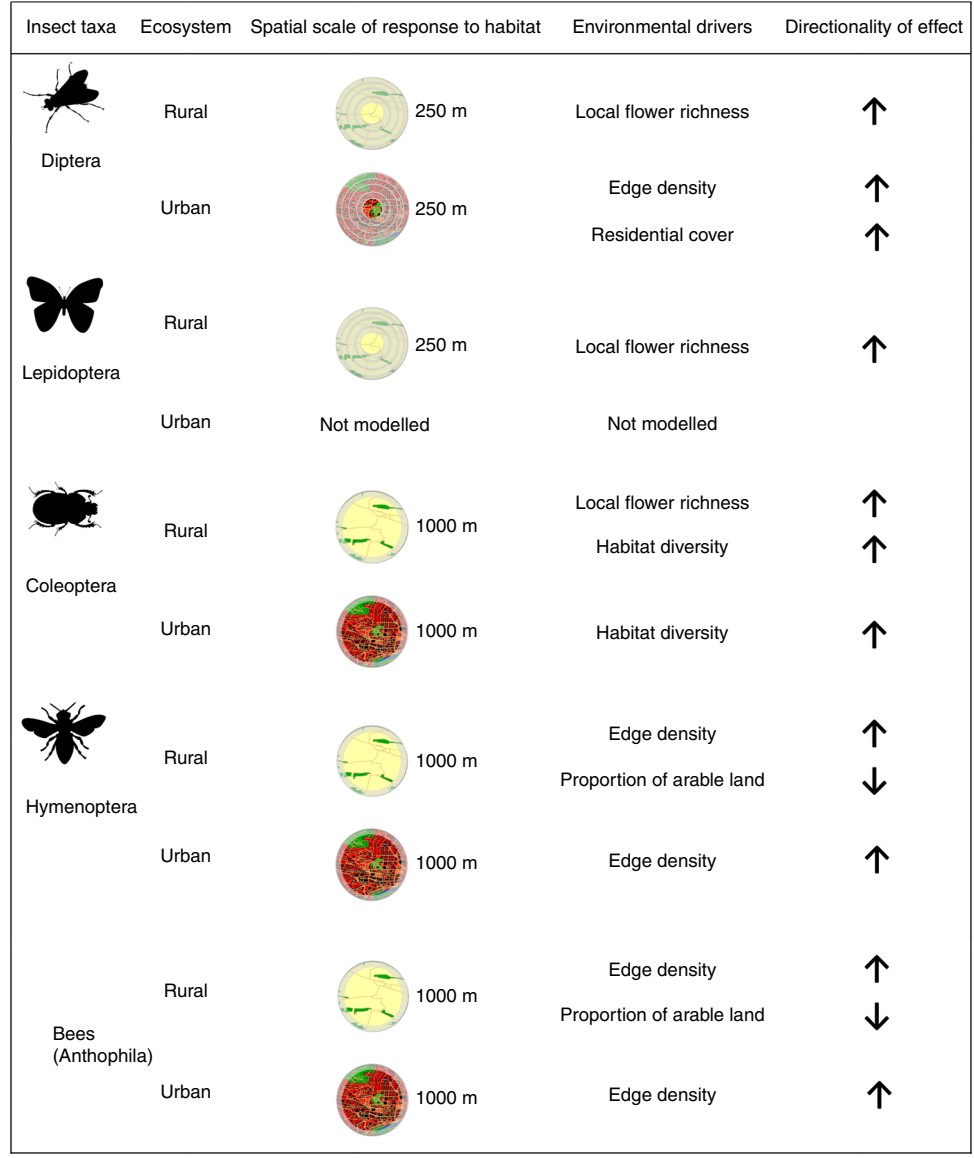

**Fig. 3 Environmental drivers of insect OTU richness in rural and urban ecosystems.** Significant (GLM; $P < 0.05$) environmental drivers of OTU richness of Diptera, Lepidoptera, Coleoptera, Hymenoptera and bees (Anthophila, a subset of Hymenoptera) in $N = 9$ rural and $N = 9$ urban sites at the local patch and landscape scales (1000 m radius for all Hymenoptera and Coleoptera, 250 m for all Diptera and Lepidoptera). Due to low sample size (mean = 1.44 ± 1.01 SD), Lepidoptera OTU richness was not modelled within urban sites. We did not find any significant predictors of hover flies (Syrphidae) OTU richness in rural or urban ecosystems. Up-arrows and down-arrows correspond to positive and negative effects, respectively. Insect silhouettes courtesy of OpenClipart-Vectors/Pixabay, Elias Schäfer/Pixabay and Clker-Free-Vector-Images/ Pixabay.

(LMM, $t = 3.888$, $P < 0.001$; Fig. 4). Though urban sites had somewhat higher flower richness than rural sites (urban: mean 13.3 plant species in flower, rural: mean 8.1 plant species in flower; GLMM, $Z = 3.350$, $P < 0.001$), urban and rural sites did not differ in the abundance of local flowering plants and in co-flowering *T. pratense* plants (LMM, $t = 0.403$, $P = 0.697$; GLMM, $Z = 0.388$, $P = 0.698$; respectively). When controlling for landscape-scale factors as well as these local patch-scale effects (namely: local flower richness, local flower abundance and the abundance of co-flowering *T. pratense* plants), seed set remained significantly higher at urban versus rural sites (LMM, $t = 2.720$, $P = 0.007$). We cannot exclude a honeypot effect having led to greater *T. pratense* pollination in urban versus rural sites. However, as described in our analysis of insect community diversity above, surrounding land use of urban and rural sites was equally inhospitable for flying insect pollinators

(Supplementary Fig. 2). The honeypot effect is likely to have operated at both urban and rural sites.

Bumble bees (*Bombus* spp.) were the dominant *T. pratense* flower visitors across all sites. During the 5400 min of direct observations of our *T. pratense* experimental plants, we observed a total of 1306 interactions ($N_{int}$) between flying insects (10 morphogroups) and red clover flowers across our 18 sites. Bumble bees (*B. lapidarius/B. sorooensis proteus*, *B. terrestris/B. lucorum*, and *B. pascuorum*, Supplementary Table 3) were involved in 75.3% ($N_{int} = 984$) of these interactions, of which *B. pascuorum* was the most prominent ($N_{int} = 714$; Supplementary Table 3). *Apis mellifera* was involved in 8.7% ($N_{int} = 114$), halictid bees in 5.1% ($N_{int} = 65$), Lepidoptera in 4.6% ($N_{int} = 60$), andrenid bees in 2.4% ($N_{int} = 31$), hover flies in 2.1% ($N_{int} = 27$), other Diptera in 1.3% ($N_{int} = 17$), and Coleoptera in 0.5% ($N_{int} = 6$) of these interactions (Supplementary Table 3).

Overall insect visitation rates to *T. pratense* plants were higher in urban compared to rural sites, even after controlling for landscape variables as well as local flower richness and local flower abundance (GLMM, $Z = 2.771$, $P = 0.005$). More specifically, visits of the two most common *T. pratense* flower visitor groups, *Bombus* spp. and *A. mellifera*, were higher in urban compared to rural sites (GLMMs: *Bombus* spp., $Z = 2.645$, $P = 0.008$, Fig. 5a; *A. mellifera*, $Z = 2.433$, $P = 0.015$, Fig. 5b). Visitation rates of all other insects did not differ between urban versus rural sites (GLMM, $Z = 1.275$, $P = 0.202$, Fig. 5c). Honey bee visitation was approximately an order of magnitude lower than that of all *Bombus* species.

Visitation rates to red clover increased only with increasing edge density in urban sites (GLM, $Z = 3.623$, $P < 0.001$) and only with increasing flower richness in rural sites (GLM, $Z = 2.969$, $P = 0.003$).

As expected, we found a positive association between visitation rates and seed set within each ecosystem as well as across all sites (LMMs, $t = 4.341$, $P < 0.001$, Fig. 6a). Within each ecosystem and

across all 18 sites, there was also an overall positive relationship between Hymenoptera OTU richness and Hymenoptera PSV with red clover seed set, when controlling for insect visitation rate (LMMs, $t = 2.294$, $P = 0.023$, Fig. 6b; $t = 5.180$, $P < 0.001$, Fig. 6c; respectively). We did not find any significant interaction effects of biodiversity (species richness, PSV) and visitation rates with ecosystem type (urban/rural; LMM; $P > 0.05$), suggesting that the relationships between insect biodiversity and red clover seed set are similar in both ecosystems.

**Disentangling the main drivers of *T. pratense* pollination.** To reveal those putatively causal factors, or interactions among them, that influenced pollination across both ecosystem types, we used structural equation modelling (SEM). Our piecewise SEM selection process yielded one final path model relating *T. pratense* seed set to overall insect visitation rates, Hymenoptera community diversity, and local patch and landscape variables, with stable fit to our data (Fisher's $C = 8.03$, d.f. = 4, $P = 0.09$; Fig. 7, Supplementary Table 4). Due to a substantial reduction in model fit, the final piecewise SEM did not include Diptera, Lepidoptera and Coleoptera community richness and several of our local patch and landscape-scale variables (e.g. the proportion of semi-natural land cover, the proportion of residential/commercial/industrial land cover, the extent of botanical gardens, public parks and allotments, conspecific donor flower density, local flower richness and abundance; Fig. 7). In the final model, overall insect visitation rates to *T. pratense* flowers and Hymenoptera species richness were positively related with edge density ($P < 0.001$; Fig. 7, Supplementary Table 4) and negatively related with the proportion of arable land in the landscape ($P < 0.001$; Fig. 7, Supplementary Table 4). Moreover, *T. pratense* plants produced more seeds with increasing insect visitation rate, increasing Hymenoptera species richness and increasing Hymenoptera PSV ($P < 0.05$, Fig. 6a–c and Fig. 7, Supplementary Table 4).

Separate SEMs of just the urban or just the rural dataset gave similar results to those of both datasets combined (Supplementary Tables 5a and b). This suggests that common ecological factors drive variation in pollination in both urban and rural ecosystems; the strength and direction of the relationship between insect biodiversity and the ecosystem service of pollination hold across urban and rural ecosystems (Fig. 7). In other words, the ecology of Hymenopteran pollinators seems not to differ between urban and rural ecosystems.

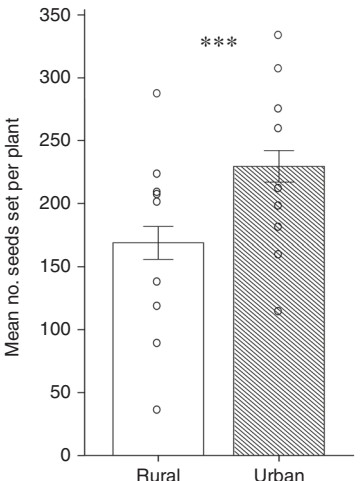

**Fig. 4 *Trifolium pratense* seed set in flower-rich rural versus paired flower-rich urban sites.** Mean number of *T. pratense* seeds set per plant ($N = 7$ inflorescences per plant, $N = 10$ plants per site) in $N = 9$ flower-rich rural versus $N = 9$ paired flower-rich urban sites; means ± SE are shown; LMM: three stars, $P < 0.001$.

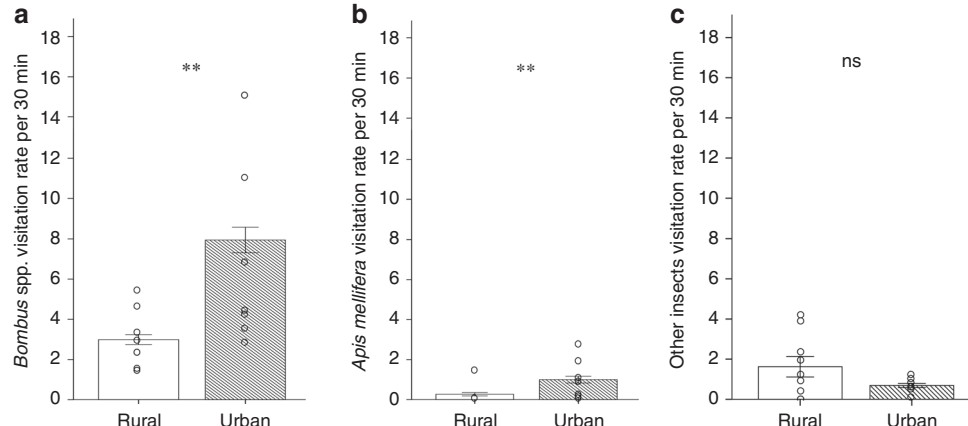

**Fig. 5 *Trifolium pratense* visitation rates in flower-rich rural versus paired flower-rich urban sites.** Flower visitation rates of **a** all *Bombus* spp. **b** *Apis mellifera* and **c** all other insects per *T. pratense* pollinometer plant per 30 min in $N = 9$ flower-rich rural versus $N = 9$ paired flower-rich urban sites; means ± SE are shown; GLMM: ns, not significant, two stars, $P < 0.01$.

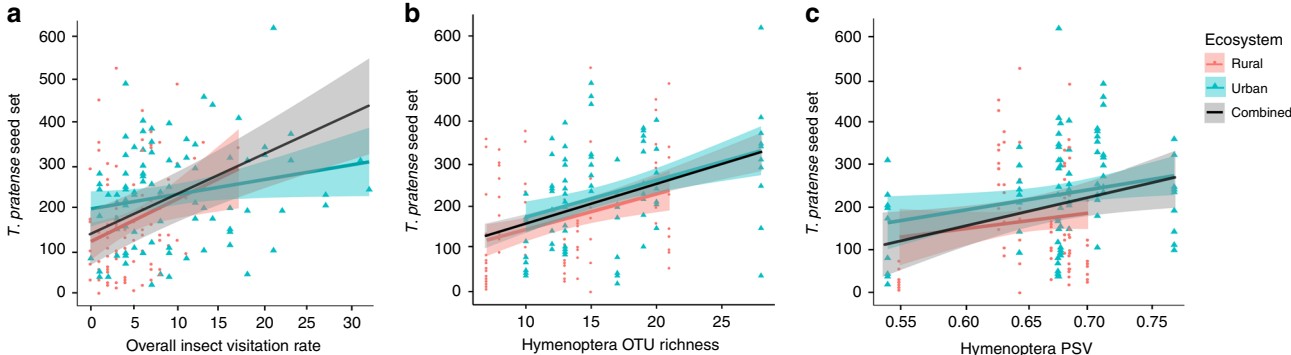

**Fig. 6 Pollinator biodiversity and *Trifolium pratense* seed set.** Relationships between *T. pratense* seed set (mean seeds per pollinometer plant) and **a** total visitation rate by all insects, **b** Hymenoptera OTU richness, and **c** Hymenoptera phylogenetic variability (PSV) across and within rural and urban sites (N = 18 sites). Plotted lines show predicted relationships and the shaded areas indicate 95% confidence intervals. Red circles and blue triangles correspond to seed set in rural and urban sites, respectively. The plotted red and blue solid lines represent the predicted relationships within the rural and urban ecosystems, respectively. The plotted black solid line represents the predicted relationship across both ecosystems, rural and urban.

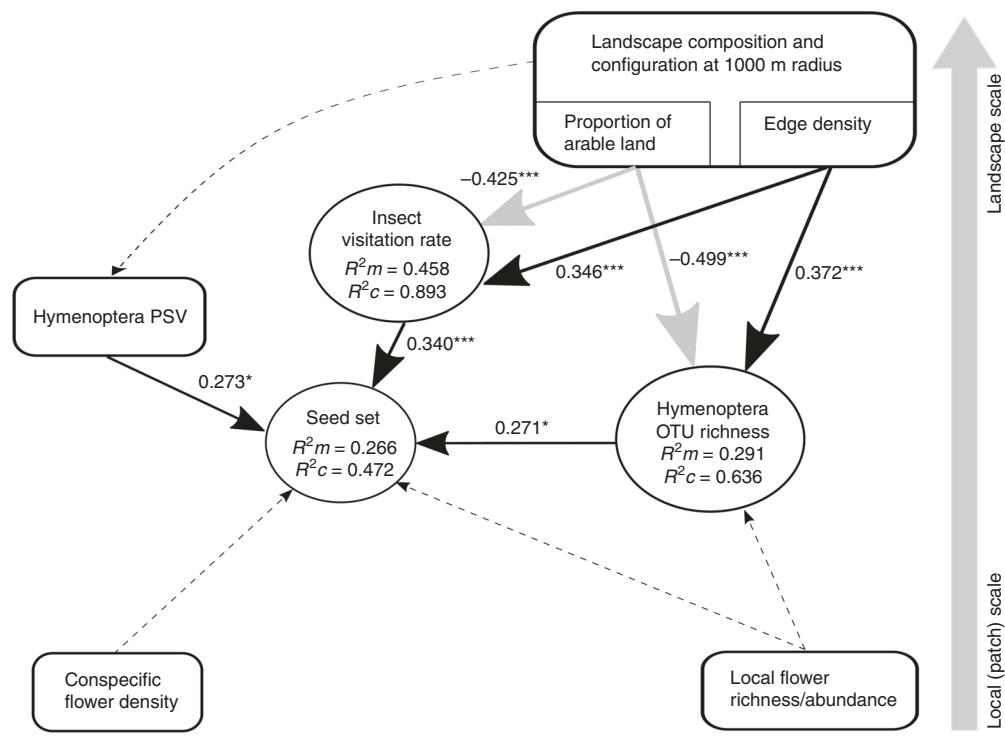

**Fig. 7 Disentangling key drivers of *Trifolium pratense* seed set across rural and urban ecosystems.** Final path model of the direct and indirect effects of flower visitation rate, pollinator diversity, local patch and landscape factors on *T. pratense* pollination in combined rural and urban sites (N = 18 sites). Black arrows show positive and grey arrows show negative effects derived from SEM analysis whilst dashed arrows show hypothesized effects. Arrow thickness corresponds to the size of path coefficients. Standardized path coefficients are reported next to the bold arrows and $R^2$ values (conditional $R^2_c$ as well as marginal $R^2_m$) are reported for all response variables within boxes; GLMM and LMM: one star, $P < 0.05$; three stars, $P < 0.001$. PSV, phylogenetic species variability. Factors potentially affecting pollination have been separated vertically by the scale at which they likely operate (local-patch to landscape scale) for clarity.

## Discussion

In this replicated study across the Central European landscape, we found that the experimentally quantified provision of pollination services was higher in urban than in rural sites. This difference was likely driven by increased visitation rates, largely of bumble bees, within urban habitats and was associated with increasing Hymenopteran phylogenetic diversity and Hymenopteran species richness, which were enhanced by increasing habitat edge density but suppressed by agriculture in the surrounding landscape. In contrast, Diptera and Lepidoptera were less diverse at urban sites.

Scale-dependent factors such as the local abundance of conspecific flowers and landscape-level habitat configurational heterogeneity (our edge density) can have a profound impact on wild bee pollinators as well as on their pollination services, measured in terms of 'quantity' (e.g. number of pollen grains removed from anthers and deposited on stigmas) and 'quality' (e.g. viability or compatibility of transported pollen grains) and seed output[36]. Yet the marked differences we found between urban versus rural insect diversity and pollination held even after controlling for differences at patch (e.g. flower richness and abundance of

conspecific flowers) and landscape (e.g. proportion of habitat suitable for pollinators in the vicinity) levels. In addition, Hymenopteran OTU richness and Hymenoptera PSV were the best predictors of pollinometer (*T. pratense*) seed set across both ecosystems. Both Hymenoptera OTU richness and insect visitation rates were suppressed by an increase in surrounding arable land use in rural sites and by decreased configurational heterogeneity in green cover (semi-natural and forest cover, botanical and public parks, and allotments) surrounding a site across both urban and rural sites, as also recently reported for rural (agricultural) sites across European nations[36]. In short, we found that urban areas acted as hotspots of pollination services for *T. pratense*, which were mainly provided by bumble bees.

Though we argue that the honeypot effect did not impact our study's response variables (insect biodiversity and pollination) because local and landscape-level ecological variables related to flying insect pollinators did not vary markedly between urban versus rural sites, we cannot formally exclude it. Replicate, landscape-level experiments selectively increasing or decreasing flower abundance and diversity might offer one option to test for its effect on insect diversity and pollination.

Though overall flying insect species richness and biomass were higher in rural compared to urban sites, patterns varied across taxonomic groups. While Diptera and Lepidoptera were more diverse in rural areas, Hymenoptera diversity was higher in urban areas. Others have also reported detrimental impacts of urbanisation on Diptera[37], Lepidoptera[38], and Coleoptera[25]. Our data suggest that Hymenoptera may be more resilient to urbanisation compared to these other three major flying insect orders, or less resilient to the rural environment dominated by agricultural land use. This may also be the case for British cities and their neighbouring rural landscapes[21].

Despite these major differences in the biodiversity of insects in urban versus rural sites, we nevertheless found similarities in their response to local and landscape variables. We found that not only Hymenoptera but also Diptera benefited from higher edge density of green cover (ecotones) in urban environments. Hymenoptera similarly benefited from higher edge density of green cover in rural environments, where they also showed a negative response to increased arable (agricultural) cover. These results importantly suggest that ecotones (edge density), rich in floral and nesting resources, might be valuable habitats for diverse pollinator groups, as also suggested for wild bees in European agricultural landscapes[36]. In addition, the positive relationship we found between edge density and pollinometer flower visitation rates further suggests that flower visitors might use ecotones as foraging routes and that they therefore could provide connectivity among suitable habitats, facilitating pollinator movement and enhancing flower visits[36,39] and plant gene flow.

We found that Coleoptera showed a positive response to landscape diversity in both urban and rural habitats whereas Lepidoptera and Diptera showed a positive response only to the availability of diverse floral resources in rural environments. As well as underscoring the negative impacts of agriculture on pollinators, these results highlight the importance of both local patch (i.e. local flower richness) and landscape heterogeneity in enhancing insect pollinator diversity[31]. They point towards the necessity of scale-dependent and taxon-specific conservation management addressing the requirements of multiple taxonomic groups to enhance overall insect diversity, both in the urban and the rural landscape.

The degree of evolutionary distinctiveness of the OTUs we sampled, measured as PSV and MNTD, did not differ between rural and urban sites. This suggests a similar degree of environmental filtering in determining the pollinator community composition in rural and urban areas. A similar lack of phylogenetic

distinctness between urban and rural areas was also reported for German vascular plants, though a higher plant richness in combination with lower than expected phylogenetic diversity in urban areas suggested strong environmental filtering of plants[14]. While overall insect species richness may be lower in urban areas, our data suggest that cities can nevertheless harbour many pollinator taxa from a diverse set of lineages.

Using a paired sampling design to compare pollination in flower-rich urban and flower-rich rural sites across multiple, independent locations and controlling for potentially confounding variables, we found higher seed set of a highly pollinator-dependent plant in cities compared to the countryside, as also found in two local studies[40,41]. Substantial evidence exists for the positive influence of biodiversity on many ecosystem services[42]. The differences in Hymenopteran, particularly bumble bee (*Bombus* spp.), biodiversity between urban and rural sites that we detected may have given rise to higher pollination in urban sites.

In our field investigation, we found that direct effects of *T. pratense* density (compatible pollen donor plants) on pollination were not significant. Rather, increased visitation rates and the richness and phylogenetic variability of local Hymenopteran communities were significantly associated with increased seed set in *T. pratense* plants. Thus, the facet of biodiversity best related to ecosystem service supply in our study was the abundance and richness of Hymenoptera, especially wild (non-managed) bees (bumble bees and other wild bee species) as well as their phylogenetic diversity, likely reflecting their ecological (functional) diversity. Recent studies support the view that pollination service provision is enhanced by high pollinator species diversity[43], including high pollinator phylogenetic diversity[35]. We nevertheless urge caution in the interpretation of our results because flowers of our pollinometer *T. pratense* plants have long corollae and were mainly visited by (long-tongued) bumble bees. Thus the differences between urban and rural sites in pollination that we recorded were likely causally related to *Bombus* visitation rates, which were higher at urban sites. Though we found Hymenoptera OTU richness to be higher in urban compared to rural areas, as also seen by other studies[21,41], and though in a previous study we found a high correlation between pollination service provision to *T. pratense* and to three other plant species, including those with open flowers[22], a more comprehensive set of pollinometer species covering diverse floral morphologies is needed to test unequivocally the role of pollinator species richness versus pollinator phylogenetic diversity in enhancing community-wide pollination.

Why were bee species diversity and their flower visitation rates higher in urban versus rural sites, despite controlling for local (patch) and landscape factors thought to influence insect pollinators and pollination service provision? The greater bee richness in our city sites compared with equivalent rural sites could reflect the diverse nesting resources (e.g. exposed soil for fossorial bees, dead wood for mason bees, and particularly cavities in walls or under buildings for bumble bees) for these pollinators in urban areas. Urban environments are often highly dynamic, and bees with their sophisticated neurosensory capacities, including well developed learning and memory, may be better suited to city life than Coleoptera, Diptera and Lepidoptera. In addition, the continuity of floral resources offered at urban sites but not at rural sites (see Baude et al.[44] for the discontinuity of floral resources in rural British habitats) may support diverse Hymenoptera communities, particularly wild bee species. Sampling across the flight season of insect pollinators is needed to test this hypothesis, though clearly it cannot apply to Coleoptera, Diptera and Lepidoptera, which were not more diverse at our urban sites. As we collected insects for only 5 days per site, intensifying sampling across the year would also ensure that the urban/rural effect we

detected in our data is robust across the phenology of temperate insects.

The other side of the coin is that Hymenoptera—especially bees—may be more sensitive to agricultural land management. This could include greater sensitivity of Hymenoptera than Coleoptera, Diptera and Lepidoptera to agricultural pesticides, leading to greater relative diversity and abundance of Hymenoptera in the urban environment. Though field studies or field-realistic doses of agricultural pesticide lead to a marked reduction in wild bee (*Bombus* spp.) fitness[45], bees appear no more sensitive than other major insect lineages to many pesticides[46]. It could also be the case that the variable urban combines all aspects that define an urban ecosystem in a synthetic variable, i.e., abiotic and biotic, including the local (patch) and landscape factors we measured (e.g. residential gardens), and which benefits Hymenoptera in comparison to modern rural ecosystems.

We found that the lower the proportion of arable land in rural sites and the higher the edge density of green cover (semi-natural and forest cover, botanical gardens and public parks, and allotments) in both urban and rural sites were major indirect ecological determinants of pollination via their positive effects on insect visitation rates and Hymenoptera OTU richness. Previous studies[20,22,31] have also found a negative impact of surrounding arable land use on Hymenoptera OTU richness. That edge density of green cover positively influenced Hymenoptera OTU richness in both of our study ecosystems (both urban and rural) and Dipteran OTU richness in urban sites supports the view that ecotones in both ecosystems could be valuable, potentially providing diverse nesting and forage resources for bees[44] as well as other flying insects. Promotion of edge habitat ought to be considered in urban as well as rural planning and management so as to foster Hymenopteran biodiversity and potentially that of other insect pollinators, too. We nevertheless urge caution in translating our results into conservation action because we also found that insect orders differed in their response to urbanisation, local and landscape ecological factors; one land management solution will not necessarily support all insect taxa. For example, measures to enhance Dipteran and Lepidopteran diversity need to acknowledge the finer scale (maximal response at the 250 m scale) to which they respond positively in comparison to Coleoptera and Hymenoptera, including bees (maximal response at the 1000 m scale).

We found honey bee flower visitation to be higher in urban versus rural sites. This most likely reflects the density and distribution of beekeepers and the current vogue in urban beekeeping[47]. Though a far less frequent visitor of *T. pratense* flowers, a higher density of *A. mellifera* at urban sites may nevertheless enhance the pollination of other wild and cultivated plants.

The decline of evolutionary distinct species is assumed to constitute an irreversible loss of function for entire ecosystems, with studies pointing to the importance of phylogenetic diversity in mediating the effects of biodiversity on ecosystem functioning[33,35]. In our study, we found phylogenetic diversity to be a valuable predictor of the ecosystem service of pollination, independently of species richness or land use. We cannot reveal the mechanism by which this effect is mediated, though a phylogenetically diverse Hymenopteran community may promote inter-flower movement and, as a consequence, pollination through functional complementarity[48]. It may be prudent to retain it in biodiversity metrics for conservation, including that of pollinators[35].

In conclusion, as cities expand worldwide, appropriate urban planning and stakeholder engagement to provide local floral resources and to increase the cover and connectivity of semi-natural vegetation and other green cover could enhance their value as refuges for species affected by agricultural intensification. Urban centres across the globe could thereby act as sources of pollinators and hotspots of the ecosystem service of pollination of urban crops and wild flowers. From the perspective of the rural landscape, similar management options are likely to enhance pollinators as well as wider flying insect diversity and support the ecosystem service of pollination for food security and wildflower reproduction.

## Methods

**Study sites.** To test the association between urban versus rural land use on insect pollinators and pollination, we used a well-replicated study design that employed a flower-rich urban site paired with a flower-rich rural site, replicated across nine major cities in central and eastern Germany. Cities were Berlin, Braunschweig, Chemnitz, Dresden, Göttingen, Halle, Jena, Leipzig and Potsdam (Fig. 1). The minimum distance between two city sites was 20 km (Fig. 1). Each adjacent rural site was a minimum of 10 km (maximum 40 km, mean 25 km; see Fig. 1) from its urban paired site and greater than 25 km from all other rural or urban sites (Fig. 1), sufficient distance for all 18 sites to be considered independent. All sites were selected to represent flower-rich habitats within the urban and rural landscapes of Central Europe (Supplementary Table 1).

Due to the well-documented relationship between pollinator diversity and flower richness[49], we chose our sites to reflect a flower-rich and pollinator-optimal comparison of the two ecosystems (Supplementary Table 1), allowing us to focus on the broader landscape context (i.e. urban versus rural ecosystems). Our logic was to compare the best urban sites for insect pollinators[15,22] with the best rural sites that were matched in terms of habitat structure (land cover, flower abundance), though each pair was sited in a different land use matrix (urban versus rural, respectively). At each site (urban and rural), we selected a 25 m × 25 m area with diverse floral resources, which we used as our urban sampling plot (Supplementary Table 1). Though sampling across a gradient of urbanisation and into the rural landscape is another sampling strategy that has been used to demonstrate the importance of cities for pollinators[19,20,22], our intention was to compare urban with rural habitats and thus we maximised the number of urban–rural comparisons for our given intensity of sampling.

Urban sites were near the urban core, surrounded by high road density and human infrastructure, and were located in or in close proximity to botanical gardens (N = 7) or public parks (N = 2). They differed in their surrounding urban matrix, though residential and commercial/industrial areas comprised a mean of 60% of the surrounding (1000 m scale) land use across all urban sites (Supplementary Table 6). Private gardens were not used for experiments due to lack of availability at all sites and access restrictions. At each site, we selected a 25 m × 25 m area with diverse floral resources, which we used as our urban sampling plot (Supplementary Table 1).

Rural sites were selected to be as similar as urban sites in terms of their local (250 m scale) land cover (i.e. flower abundance) by using land cover maps within a Geographic Information System (Quantum QIS). Arable land (=agricultural land) and forest/semi-natural cover were the dominant land use types at the landscape scale, comprising a mean of 45% and 41% the of surrounding (1000 m scale) land use across all rural sites, respectively (Supplementary Table 7), typical of the region's rural environment i.e. our rural sites were not impacted by urban sprawl. To identify rural sites that were ideal for insect pollinators yet independent of urban sites, we drew a buffer at a circumference of 10 km radius from each urban site and then used GIS to identify areas of semi-natural grassland and scrub vegetation immediately outside the buffer that were largely devoid of 'residential' and 'commercial/industrial' and dominated by arable land and forest/semi-natural cover within the surrounding 1 km radius. Candidate rural sites were then visited and one was randomly selected that had a 25 m × 25 m area near its geographical centre with diverse floral resources (Supplementary Table 1), which we used as our rural sampling plot. By using these criteria for site selection, we aimed to sample from the best sites for insect pollinators, and potentially also for pollination, among urban and among rural localities.

**Measurement of flying insect diversity.** To compare the diversity of flying insects at urban flower-rich sites with those at rural flower-rich sites, we measured species richness of the four main orders of flying insect pollinators: Diptera, Lepidoptera, Coleoptera, and Hymenoptera[50]. Though our experimental *Trifolium pratense* plants were primarily visited by bees (see above), our attention to all four orders allows us to address the pollinators of a broader range of plant species, and to explore their taxonomic diversity in response to urbanisation. We employed pan traps to sample insects, which are a standardised and commonly used method for collecting flying insects that have been widely used in studies of pollinator communities[51,52]. As for any other flying insect sampling method, pan traps have disadvantages such as potential taxonomic bias and under-sampling of large insects. However, in a comparative study, they were found to be the most efficient method of sampling bees across geographic regions in Europe[51]. Additionally, coloured pan traps have been found to outperform malaise traps in sampling all major insect pollinators, including Diptera, Lepidoptera and Coleoptera[52].

Each urban–rural site pair was visited at the same time and for a total of five consecutive warm, non-windy days at one point between June and August 2014 (Supplementary Table 8). Temperatures exceeded 16 °C, wind speed was less than 2 m/s at 1 m above ground level, and skies were sunny (<50% cloud cover; Supplementary Table 8) on all sampling days. Insects were sampled using three blue, three yellow and three white pan traps (diameter: 42 cm, height: 2.8 cm) mounted on a stick at vegetation height at each site and set out across the 25 m × 25 m plot. Each pan trap was 2/3 filled with unscented soapy water and emptied every day. Insects from traps were stored in 95% ethanol. Insect samples from each site were washed, dried and weighed using a balance (Denver Instruments SI-2002A, Denver, USA).

For assessment of species richness, we used next generation sequencing (NGS)-based metabarcoding[53] to identify the number of different species of Diptera, Lepidoptera, Coleoptera, and Hymenoptera based on mitochondrial COI DNA sequences. Metabarcoding has been successfully used to assess patterns of arthropod diversity, and has proven faster, cheaper and more efficient than traditional morphological taxonomy[53,54]. This is likely to be particularly the case for the large and diverse order Diptera[55]. A disadvantage of this approach is that it provides only a list of species sampled at a site, not their relative abundance, and it may have under-recorded small or rare species that occurred as singletons in our pan trap material. Our metabarcoding protocol combined mass-PCR amplification of the COI barcode gene, 454-pyrosequencing and several optimised bioinformatics steps to determine OTUs and perform taxonomic assignment to species (Supplementary Methods).

In brief, denoised 454-pyrosequencing reads of pan-trap material from all 18 sites were reduced 1.8-fold from 157,310 raw reads to 85,223 Insecta-only sequences (orders: Diptera, Lepidoptera, Coleoptera and Hymenoptera). After removal of singletons, our total metabarcoding dataset contained 592 Insecta OTUs. Of these, 342 (57.8%) belonged to Diptera, 116 (19.6%) to Hymenoptera, 81 (13.7%) to Lepidoptera and 53 (9.0%) to Coleoptera (the main insect orders sampled; see Supplementary Fig. 4). The majority of OTUs (308 out of 592, 66.2%) were successfully assigned to species level (Supplementary Dataset). For Diptera we could assign 117 OTUs (34.2%); for Hymenoptera 85 OTUs (73.2%); for Lepidoptera 65 OTUs (80.2%) and for Coleoptera 41 OTUs (77.3%) to species (Supplementary Dataset), a rate which is typical for Central European barcoding studies[55]. We henceforth use species richness and OTU richness interchangeably. The number of reads was not correlated with OTU richness across our dataset (Pearson's product-moment correlation $r = 0.033$, $N = 18$, $P > 0.05$), suggesting we had sufficient reads to saturate our OTU assessment per site.

To explore whether phylogenetic diversity and, by inference, trait diversity[56] of flying insects were related to the ecosystem service of pollination and how they were impacted by urbanisation, we additionally estimated two phylogenetic diversity metrics, (i) PSV and (ii) MNTD, using the R package *picante*[57]. PSV quantifies how phylogenetic relatedness decreases the variance of a hypothetical unselected/neutral trait shared by all species in a community. The expected value of PSV is statistically independent of species richness; it is 1 when all species in a sample are unrelated and approaches 0 as species become more related[58]. MNTD provides an average of the distances between each species and its nearest phylogenetic neighbour in the community. MNTD quantifies the degree that a community may be a set of closely related species versus a heterogeneous set of taxa from disparate taxonomic clades[59]. It was only weakly related to PSV in our dataset (Pearson's product-moment correlation $r = 0.441$, $N = 18$, $P = 0.066$) and therefore we retained it in analyses as a second measure of phylogenetic diversity.

**Quantifying the ecosystem service of pollination.** To quantify differences in pollination service provision in urban versus rural ecosystems, we used greenhouse-raised *Trifolium pratense* plants as 'pollinometers' at each site and evaluated their pollination success. Though *T. pratense* is preferentially visited by long-tongued bumble bees[22,60], in a previous study[22,41] we found high correlations in the pollination success of *Trifolium pratense, Borago officinalis, Sinapis alba* and *Trifolium repens* experimental plants (common flowering plants in Germany; results in Supplementary Fig. 5 and Supplementary Table 9), suggesting that *T. pratense* could be used as a model system to quantify the ecosystem service of pollination in our study region.

Seeds of wild *T. pratense* were obtained from a local seed provider (Rieger-Hofmann GmbH, Blaufelden, Germany). These (diploid) wild plants were readily visited by bumble bees (*Bombus* spp.), honey bees (*Apis mellifera*) and other wild bees (other Anthophila) (see results). All seeds were germinated and grown for two months in an insect-free glasshouse before placement at our study sites. Ten potted plants, each with eight open inflorescences (flower heads) ($N = 8$ per plant) marked with coloured tape, were placed at each field site during the five flying insect sampling days at that site. In each plant, one inflorescence (entire flower head containing ca. 100 individual flowers) of the eight was bagged throughout the experiment in the field with fine net (1 mm gauze) to prevent visitation by pollinators (zero control). Bagged flowers did not set any seed, demonstrating the dependence of *T. pratense* on insect visitation for seed set.

Pollinometer plants in each community were randomly ordered at one metre distance along a transect of 10 m × 1 m within the 25 m × 25 m plot at each site. Once the 5 days of the pollination experiment were completed at a site, focal plants were returned to the insect-free greenhouse until seeds were formed. Seeds from all

seven unbagged inflorescences per plant were counted and the average number of seeds per plant (i.e. per seven inflorescences) was used as a measure of the ecosystem service of pollination. Each inflorescence of *T. pratense* contains up to 100 flowers. We weighed all marked, dried inflorescences per plant as a surrogate of the number of flowers in each inflorescence and tested for differences in flower number per inflorescence between the two ecosystems. We did not find any differences in the weight of inflorescences between the two ecosystems (LMM, $t = 0.183$, $P = 0.855$), suggesting that an equivalent number of flowers had been exposed to pollinators at each member of a site pair. Thus, our method of assessment of the ecosystem service of pollination did not differ between an urban-rural pair of sites and is presented in terms of number of seeds per plant (i.e. per seven inflorescences per plant exposed to pollinators).

In addition, we monitored all insects visiting the flowers of experimental *T. pratense* plants for five hours at each site in order to estimate flower visitation rates because these are causally related to pollination success. Due to the possible confounding effect of flower visitation frequency when testing the effect of pollinator diversity on pollination[61], we used flower visitation rate as a covariate in our statistical analysis of the relationship between pollinator diversity and pollination. Each experimental plant was observed twice per site (15 min in the morning and 15 min in the afternoon), for a total of 300 min observation time of *T. pratense* pollinometers per site. Visitor identity (11 morphogroups: Coleoptera; Syrphidae; other Diptera; Lepidoptera; wasps; bees of each family: Andrenidae, Halictidae; plus each morphospecies group: *Bombus lapidarius/Bombus sorooensis proteus*; *Bombus terrestris/Bombus lucorum*; *Bombus pascuorum*; and *Apis mellifera*) was recorded. Furthermore, at each site we estimated the abundance of conspecific pollen donors by counting the number of inflorescences of co-flowering *T. pratense* plants within a 200 m buffer around each plot (Supplementary Table 1).

**Local patch and landscape variables.** To determine the main ecological correlates of insect biodiversity and pollination in both rural and urban flower-rich sites, we gathered a series of local (patch) and landscape-scale variables potentially related to insect pollinators and pollination. Although we selected sites with high availability of floral resources in both ecosystems, we additionally quantified local flowering plant richness and abundance (number of flowers of each plant species) as an estimator of floral resource availability using 10 randomly placed 1 m$^2$ quadrats at each of our sampling sites (Supplementary Table 1). These estimates were used in our models as covariates. We also quantified habitat composition at five spatial scales (250, 500, 750, 1000 and 1500 m) per site using Quantum GIS and land cover data obtained from Geofabrik GmbH (http://www.geofabrik.de/). We calculated landscape diversity for each radius using the Shannon–Wiener index ($H_s$): $H_s = -\Sigma p_i \bullet \ln p_i$, where $p_i$ is the proportion of each land cover of type $i$, as defined in Supplementary Table 10.

To identify the scale at which surrounding land cover had the most power to explain each insect order's occurrence, we correlated each insect order's OTU richness with landscape diversity (measured as Shannon–Weiner diversity of proportions of land use types) at each of our study sites at all five scales[62]. Correlation coefficients peaked at the 250 m scale for Diptera and Lepidoptera OTU richness and at the 1000 m scale for Coleoptera and Hymenoptera OTU richness both across (combining urban and rural sites in one analysis) and within each ecosystem (analysing urban and rural sites separately) (Supplementary Table 11). These spatial scales for each taxon were then used for subsequent landscape-scale analyses.

Several metrics known to impact flying insect biodiversity were used to quantify landscape heterogeneity (composition and configuration) at each of the 18 sites at both 250 and 1000 m scales[28,63]. These were (i) the proportion of semi-natural cover (grassland, meadows and scrub vegetation), (ii) the proportion of forest, (iii) the extent of arable (=agricultural) cover, (iv) the proportion of residential and (v) commercial/industrial land cover, (vi) the extent of botanical gardens, public parks and allotments, and (vii) edge density, as total length of 'green cover' (semi-natural and forest cover, botanical gardens, public parks, and allotments) patch edges divided by the total area, and which represents a quantification of landscape configuration.

**Statistical analyses.** Given our paired 'urban-rural' experimental design, the rationale in our statistical analyses was to use site pair as a random factor and to compare between ecosystem type (urban versus rural) because sites had been selected a priori as belonging to either urban or rural ecosystems. We controlled for potentially confounding local and landscape factors, unless we specifically aimed to model their relationship to predictor variables: dimensions of biodiversity and pollination. Results reported below were qualitatively the same in all analyses when we did not control for potentially confounding local and landscape variables, supporting the view that we had selected equivalent sites in urban and rural ecosystems and reinforcing the robustness of our results.

To derive comparable estimates across urban and rural sites, we standardized all quantitative predictors to a mean of zero and standard deviation of one. Prior to each analysis we used variance inflation factors (VIFs) to check for collinearity among our explanatory variables. VIFs were lower than 3 for all predictors in all models tested, indicating negligible levels of collinearity[64].

To determine whether flower-rich urban sites supported a higher insect pollinator biodiversity than flower-rich rural sites, we tested for differences in

flying insect species richness between rural and urban sites using generalised linear mixed models (GLMMs) with Poisson error structure and site pair as a random effect factor. We undertook these analyses for all insects and for each insect order separately, using all local patch (i.e. flower richness and abundance) and landscape variables (i.e. proportion of semi-natural cover, forest, arable, residential, commercial/industrial, botanical gardens/parks/allotments, habitat diversity and edge density) as covariates. For the Diptera and Hymenoptera datasets, we additionally analysed subsets of each, namely the hover flies (Diptera: Syrphidae) and the bees (Hymenoptera: Anthophila), as they are considered as the most important insect pollinator taxa within their respective orders[50]. We also tested for differences in flying insect biomass (log of weight in g) between urban and rural sites using a linear mixed model (LMM). To test for differences between rural and urban flower-rich sites with respect to our phylogenetic diversity metrics, we used LMMs for MNTD and GLMMs with binomial error structure for PSV. Again, site pair was included as a random effect factor and all local patch and landscape-scale factors described above were included as covariates.

Flower-rich sites located within an inhospitable landscape may attract insects from further afield than sites nested within a floristically rich landscape, a 'honeypot' effect. Though we did not quantify floral abundance across the wider landscape so as to test explicitly for the honeypot effect, we tested for differences between urban versus rural sites in terms of local (flower abundance, flower diversity) and landscape variables (green cover, edge density) using LMMs and GLMMs to assess whether they varied consistently between ecosystems.

To analyse the effects of local (patch) habitat (i.e. flower richness and abundance) and landscape composition (i.e. proportion of semi-natural cover, forest, arable, residential, commercial/industrial and botanical gardens/parks/allotments and habitat diversity) and configuration (i.e. edge density) on species diversity for each insect order, we used generalised linear models (GLMs) with Julian day as a covariate. We explored these drivers of insect diversity for rural and urban ecosystems separately because sites had been selected a priori to represent flower-rich locations typical of each ecosystem but with stark differences between ecosystem types (urban: mean 60% residential/industrial land cover; rural: mean 86% agricultural/forest/semi-natural land cover). For each insect order, we used the spatial scale derived from their response to landscape diversity (for Diptera and Lepidoptera, 250 m; for Coleoptera and Hymenoptera, 1000 m, see the "Results" section) and Julian day was used as a covariate.

To test for differences in insect community composition between flower-rich urban and rural sites, we performed a paired permutational multivariate analysis of variance using the *adonis* function, with 999 permutations, implemented in the R package *vegan*[65]. In the *adonis* analysis, the Jaccard distance matrix of species composition was the response variable, with ecosystem (urban/rural) as a fixed factor. The strata (block) argument was set to 'site pair' so that randomizations were constrained to occur within each pair and not across all sample sites. We undertook these analyses for all insects and for Hymenoptera only. We employed non-metric multidimensional scaling (NMDS) within the package *vegan* to visualize the variation in community composition. For each site we also calculated the mean ecological distance (Jaccard index) over all pairwise comparisons of the 9 sites belonging to the same ecosystem type. We used a LMM to compare urban and rural ecosystems, with pair as a random effect factor and using all local patch and landscape variables as covariates.

We tested the effects of each ecosystem (urban versus rural) on the pollination of *T. pratense* plants using LMMs. In doing so, we included local flower richness, local flower abundance, abundance of co-flowering *T. pratense* plants and landscape-scale factors (i.e. proportion of semi-natural cover, forest, arable, botanical gardens/public parks/allotments, habitat diversity and edge density) as covariates. Individual plants were nested within each site and site pair was used as a random effect.

To explore the main environmental correlates of visitation rates in each ecosystem (urban, rural), we used GLMs, exactly as we did when testing the main environmental correlates of insect species diversity (i.e. for each ecosystem separately).

We then used LMMs to explore the relationships between insect biodiversity and *T. pratense* seed set while controlling for flower visitation rates. In doing so, site pair was used as a random effect and ecosystem (urban versus rural) as a fixed effect factor. We modelled seed set as dependent on insect species richness, PSV and MNTD, measured over all Insecta and for each insect order separately.

When investigating the main environmental drivers of insect species diversity, we used an all-subset (i.e. all combinations of predictors of interest) automated model selection approach based on the Akaike Information Criterion, corrected for small sample size (AICc), using the *dredge* function (R package *MuMIn*[66]) and with a maximum of three predictors to avoid overfitting. All mixed model analyses were performed using the R package *lme4*[67]. All model (GLMMs, LMMs and GLMs) assumptions were checked visually. The residuals of all regression models were tested for spatial and temporal autocorrelation using Moran's *I* implemented in the R package *ape*[68]. Residuals were not found to be autocorrelated ($P > 0.05$).

Finally, having identified differences in pollinator diversity and pollination service provision at urban versus rural sites, and the local and landscape factors potentially driving urban-rural differences, we synthesised our analyses by

exploring commonalities across all (urban and rural) sites. Here our aim was to identify the most important putative causes of variation in pollination, regardless of ecosystem type, and hence we did not use ecosystem type (urban/rural) as a fixed factor in analyses, though we did retain pair as a random effect factor to account for our experimental design in which pairs of sites were investigated sequentially across the summer. To do so, we used piecewise SEM, which allowed us to visualise and also statistically test for the importance of factors and their interrelations in a logical, causal path. In keeping with the main rationale of our study, we also used SEMs to identify the main factors associated with pollinator biodiversity and pollination in rural versus urban sites by running separate SEMs for each ecosystem. The similarity of SEMs for each ecosystem (Supplementary Tables 8a and b), dominated by the predictors: edge density, Hymenoptera species richness and Hymenoptera phylogenetic diversity as well as plant visitation rate, justify our synthetic approach of exploring factors affecting pollination across both ecosystems within one SEM.

In generating all SEMs, we hypothesised that landscape composition and configuration as well as local patch flower richness and abundance, and conspecific pollen donor availability might have affected *T. pratense* seed set directly and also indirectly through affecting insect visitation rates, OTU richness and phylogenetic diversity (PSV and MNTD; data for Diptera, Lepidoptera, Coleoptera and Hymenoptera added separately). We performed piecewise SEM analyses using the R package *piecewiseSEM*[69]. From an overall model based on a priori knowledge of interactions with all hypothesised effects, we used a backwards stepwise elimination process based on AICc to remove non-significant pathways. In addition, we used the d-separation (d-sep) test to evaluate whether the non-hypothesised, independent paths were significant and whether the models incorporated into the SEM could be improved by the inclusion of any of the missing path(s). We assessed goodness of fit of the final model using the Fisher's C statistic. Path coefficients and deviance explained were then calculated for each model. We report both conditional ($R^2_c$, all factors) and marginal ($R^2_m$, fixed factors only) coefficients of determination for generalized linear mixed effect models incorporated in the SEM.

All statistical analyses were performed in R v. 3.5.2[70].

**Reporting summary**. Further information on research design is available in the Nature Research Reporting Summary linked to this article.

## Data availability
Demultiplexed, raw 454-pyrosequencing reads are available under the accession number SRP096003 at the NCBI Sequence Read Archive database. Bioinformatics analysis script used for the metabarcoding and the distribution of OTUs across sites are available in the figshare Digital Repository [https://doi.org/10.6084/m9.figshare.10304795.v1].

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

## Acknowledgements

We thank Matthias Seidel, Patricia Landaverde and Lara-Sophie Dey for their assistance in the greenhouse and during fieldwork as well as Mark Frenzel, Ines Merbach and the technicians of the UFZ experimental station Bad Lauchstädt, who helped to grow and maintain the experimental plants. We would also like to thank Beatrix Schnabel for assisting with DNA library preparation for metabarcoding. We additionally thank taxonomic experts Martin Musche, UFZ (Diptera), and Matthias Seidel, Charles University Prague (Coleoptera), for their assistance in checking the list of species assigned to each of the OTUs in our dataset. Finally, we thank Prof. Douglas W. Yu for providing the raw datasets used to test our bioinformatics pipeline. We gratefully acknowledge financial support of the German Centre for Integrative Biodiversity Research (iDiv) Halle-Jena-Leipzig (Flexpool project 50170649), funded by the German Research Foundation (DFG: FZT 118).

## Author contributions

P.T. participated in the design of the study, collected field data, carried out the molecular lab work, undertook data analysis and drafted the manuscript; R.R. collected field data; G.L. undertook metabarcoding data analysis; B.K. assisted in data analysis; J.S., O.S., T.W., C.B., I.G. and M.H. discussed the concept and implementation of the study; T.E.M. and R.J.P. participated in study design and data interpretation. All authors contributed in drafting and finalising the manuscript.

## Competing interests

The authors declare no competing interests.
