## [Peer Review File · Nature Communications]

Reviewers' Comments:

Reviewer #4:

Remarks to the Author:

So firstly I understand that this has been redirected from its originally intended journals and I can see it has undergone significant review during this process. I can clearly see this rigorous review process reflected in the paper I am now reviewing. It's something of a beast of a paper, its pretty bit and part of me wonders whether cutting it down in size would not be a good idea, however, that's a call for you or the editor. I don't really mind on this front. In general I think its solidly written and analysed, although I have a few issues relating to some of the premises. In particular if you can justify /caveat some of my concerns that would be great. Personally I don't really buy into the wonders of the urban environment for conservation, however, that's what data is for and its good you fill this hole. As you can see below I just want it to be clear what the limitations are in relation to what you have done. As I say bit ones are: 1) who is benefiting from ES in cities (described in detail below); 2) limitations of the Trifolium phytometer; 3) limitations of choosing your 'rural' site as a comparison; 4) honeypot issues where bees are being concentrated into small patches in urban environment. To be fair many of these you do address in part, but humour a clinic with some more clarification and ill be happy.

Abstract

L49 'that they provide'

L50: Are you talking about specific species (ie.. *B. terrestris* in rural vs urban, or a more general thing. To be fair I know this is a short word count for the abstract so clarification is likely to be limited, but it's a bit unclear.

L56 , but

L56: So full disclosure ive never really bought into this urban pollination argument as it seems a bit contrived to me. That asside, when you talk about pollination services its got to be related to something, crops are an obvious one, but wild plants are clearly another (and I would go out of my way to argue here that it needs to be native rather than non native plants which ultimately represent potentially invasive species). So to justify the 'increases pollination services line' I think it needs to be clear here who benefits. If there are either 1) lots of crops in cities (which in general their aren't) or 2) lots of native plant species, ideally rare ones (again I am not clear that there are, but there may be) this would be a justification of why cities are so great. I would like this clarification, justification or caveat made clear somewhere if you haven't already don't this, and while it would be nice in the abstract I do see space as a problem.

Introduction

L70: Ok, so the biodiversity thing is one thing, but I would really like to see some clear justification of the pollination ES component in relation to cities. See points above. Note simply saying the may spill over to me would need some convincing given their foraging ranges would mean that from a large conurbation this would only be say a couple of km around the edges – not necessarily relevant for the countryside in general (say in comparison to a spread of semi-nat habitat though the countryside). I am sure you can put an argument together. Ultimately I take your point on the L72 that a lot of agricultural land is shit for pollinators, and that there is sig potential for urban landscape to support diversity, but ultimately bees in agriculture landscape are providing crop pollination services. Again if you can show a prevalence of native species of plant in cities then the pollination element here is more justifiable.

L80 – Ok so expand on this – if you are talking about city gardens like they have in Cuba, I get it. If its vertical farming that's more insulated from wild pollination (more likely to use greenhouse pollinator systems like *B. terrestris* colonies).

L120; Isn't it used more though as a justification for why we focus on phylogenetically distinct species like gorillas as (where losing a species is significant) compared to say butterflies (where we lose species every day and no one really cares. To be fair, it's a matter of scale I suspect.

L132-139: some of this detail seems more suited to the methods (possibly you were asked by another reviewer to put it in there).

Methods

L148: Do you really need to say 'statistically powerful' – its just a blocked design, your not reinventing the wheel or anything.

L159: I could do with some more description of these rural sites, as at the end of the day the comparison lives or dies on the basis of the sites you chose. I think giving some more clear criteria (even if the detail is in an appendix) would be useful here. Ultimately a different set of criteria for the rural sites could have yielded very different results. I don't have an issue with your design, but that caveat would be good. It also crosses my mind you may want to justify the benefits of your approach over or pros/cons of the obvious alternative design of an urbanisation gradient.

L166-168: So maybe you talk about this later, but if not I would like you to do this, however it strikes me you have a honeypot situations going on here. Do you have a high (phylogenetic) diversity, or is it at a sampling effect of what bees there are being concentrated into a small patch of limited resources. It the difference between quantifying diversity in a 2 x 2m quadrat in a field of oilseed rape vs. the same area of oilseed rape on its own in an area otherwise relatively impoverished of flowering plants (e.g. a 2 x 2 m area of OSR in a wheat field). Its likely you may see only a handful of bees in the quadrat in the OSR field, but many more species in the isolated patch as they are focusing in on this resources from a large area. Again, doesn't negate what you have done, but it seems a potentially confounding factor.

L171: so forest / semi-nat covers a pretty broad brush of habitats. Within this spectrum I would have thought say species rich grassland dominated would be better for bees than forest dominated. Was that an issue?

L199-202: 5 consecutive days of sampling – all in one - seem problematic to me. What about temporal variability across the year. I would have preferred say 3 days of sampling, but separated by a month each. Do you think this will impact the results?

L237: give a reference for the link between phyl and trait diversity. If I am honest if you are explicitly just measuring phyl diversity I would just refer to this, and maybe in the discussion mention the two are correlated. If you are explicitly deriving trait diversity fair enough.

L254: So I have already expressed my feeling that you need to justify the Es approach more. Its also clear that you have picked a flower that has a structure that precludes shorter tongued species. I accept you cant do everything, but again this should be made clear that this is not necessarily the best phytometer for many species, who may be excluded from using it. Flies jump to mind here as being obvious ones, but I would have thought that shorter tongued solitaries would also not really pollinate this. I get your justification with the correlations, which goes somewhat to justify this (L258) but presumably this just means there were lots of *Bombus* in all communities rather than it's a good model for all species.

L275: So my point above about the honeypot issue in cities (i.e. an isolated patch of flowers attracting in lots of bees (including honeybees which are often kept in cities and will really narrow in on these areas) to a small patch in a sea of concrete compared to the diluting effect of a rural landscape with lots of flowers is also very relevant here for the phytometers. How do you deal with this problem – I would have thought it really causes potentially large problems with this methods. If you limited it to virgin flowers and say X visitations by pollinators before bagging, at least this would be standardised. That said that sounds horrific to do, so maybe just caveat the issue here so its clear this is trends may

be a potential artefact.

L290 – Ok good, in relation to my above comment you have considered this -I would still explain the potential risks linked to this though.

L320-327: Fair enough, it's a reasonable approach to get a sensible scale for you landscapes.

L421: So my understanding (and correct me if I am wrong) was that structural equation models required huge numbers of replicates to produce anything reliable, you don't actually seem to have that in your study. I am guessing their are metrics for describing the reliability of model outputs, it would be good to explain how you are using them here. Is this what L439 is doing?

Results

Happy with these

Discussion

So I am happy with the discussion – what I want is the limitations of what you have done to be made clear. I can see this has on=undergone extensive review and while I don't want you to start from scratch I do see some of the approaches as having the potential to create artefacts. As long as these limitations are clear this works for me.

L564: this may be overplaying it, its 9 cities in Germany. Its undeniably a C.European landscape but this would suggest a cross country scope of the study sites to me.

Reviewer #5:

Remarks to the Author:

I am primarily providing a technical evaluation of the metabarcoding methods in the revised manuscript by Theodorou et al "Flower-rich urban areas can act as hotspots for bees and the ecosystem service of pollination but are not a panacea for all insects"

METABARCODING METHODS (BASED ON VEIWINING OF THE REVISED MANUSCRIPT ONLY)

The metabarcoding approach used is a bit dated (large PCR fragments, 454 technology), but methods are done to a standard that has been common in the literature. The use of only 18 samples (one per site Line 122 supporting material) is unfortunate. Insects were sampled on 5 consecutive days and each day could have been analysed separately (90 samples) to give extra insight.

There are also few pieces of data useful for interpretation that are missing:

- The final number of sequence reads per sample and rarefaction curves should be reported in the supporting material. The reported lack of correlation between read number and diversity (line 234) seems to be hand waving.
- The bioinformatics pipeline was evaluated with data from another exemplar study, and for what it's worth the results appear robust. For completeness it is also important to report the number of OTUs recovered that didn't match the reference Sanger OTUs and number of recovered OTUs that matched the same reference species.

Regardless of potential technical issues, the same metabarcoding methods were being applied across the sites, so even if there are some biases, the comparisons of alpha diversity should be valid.

While the metabarcoding approach is good, the study would have been considerably stronger in my mind if supported by some morphological identification and some measure of insect group relative abundance. The real strength of metabarcoding applications is in systems that are very diverse, or where identification isn't feasible – this doesn't seem to be true in this study.

An additional analysis using the current dataset that would strengthen the manuscript is inclusion of some measures of beta-diversity. Are the urban communities more similar to each other, or to the communities in the neighbouring rural sites? Are the same hymenopteran species being boosted in all the urban centres?

The experiment represents a considerable amount of work, careful analysis and presentation. The relatively limited geographical scale of the study has mixed impact. It may have strengthened the conclusions – since the urban and rural landscapes being compared are likely to have a similar signal. However, this does necessarily constrain the generality of the findings (i.e. do the findings hold outside of Europe).

COMMENTS ON RESPONSES TO PREVIOUS REVIEWER COMMENTS

Two of the weaknesses I raised above (the value morphological identification would have added; limitation on conclusions due to the relatively small geographic scale) were raised previously, but the authors can't make changes to improve these aspects of the study.

The technical responses to Reviewer 2 on metabarcoding seem reasonable. As outlined above the methods used in the study are slightly dated, but the analysis performed here are defensible. Several aspects of data analysis were improved in response to the reviewers' comments. I agree with the authors' sentiment that there are ongoing debates regarding bioinformatics and OTU clustering, and this fine-tuning will be unlikely to affect their comparisons of species diversity. Looking at iNEXT for rarefaction curves would be useful even if the authors don't extrapolate species richness.

The metabarcoding component of the paper is suitable for publication, but it does not strengthen the paper. Comments on the significance of the main conclusions are best left to other reviewers.

Reviewers' comments:

Reviewer #4 (Remarks to the Author):

Comment: So firstly I understand that this has been redirected from its originally intended journals and I can see it has undergone significant review during this process. I can clearly see this rigorous review process reflected in the paper I am now reviewing. It's something of a beast of a paper, it's pretty big and part of me wonders whether cutting it down in size would not be a good idea, however, that's a call for you or the editor. I don't really mind on this front. In general I think its solidly written and analysed, although I have a few issues relating to some of the premises. In particular if you can justify /caveat some of my concerns that would be great. Personally I don't really buy into the wonders of the urban environment for conservation, however, that's what data is for and its good you fill this hole. As you can see below I just want it to be clear what the limitations are in relation to what you have done. As I say bit ones are: 1) who is benefiting from ES in cities (described in detail below); 2) limitations of the Trifolium phytometer; 3) limitations of choosing your 'rural' site as a comparison; 4) honeypot issues where bees are being concentrated into small patches in urban environment. To be fair many of these you do address in part, but humour a cynic with some more clarification and I'll be happy.

Reply: We thank the referee for their positive comments and their four specific points of critique, to which we respond as each arises in the specific comments (below). In relation to the first issue about the size of the manuscript, we had considered splitting it into two, one addressing 'just bees' and one dealing with 'all other insects'. But, if we had done so, we felt we would have generated an overly simplistic view that has been often made (that bees fare well in cities compared to the countryside) whilst missing out on the bigger picture, that bees might be doing OK in the urban but that other insect taxa are not. For information, we submitted the ms originally to Nature: Ecology&Evolution, from which it was rejected after the first set of reviews (responses to which you presumably saw). We nevertheless revised the ms, reduced it considerably in length (deleting a RADseq-based population genetic analysis of *Bombus lapidarius*), and resubmitted it to Nature:E&E. The Nature:EE editor did not accept our request for re-review but suggested we transfer the submission to Nature Communications, which is where we now are.

Abstract

Comment: L49 'that they provide'

Reply: Removed. L50-51. "*cities impact flying insects and the ecosystem service of pollination they provide*".

Comment: L50: Are you talking about specific species (i.e. *B. terrestris* in rural vs urban, or a more general thing. To be fair I know this is a short word count for the abstract so clarification is likely to be limited, but it's a bit unclear.

Reply: We were referring to the ES of pollination provided by flying insects but,

seeing the potential for confusion, we clarified the text (L59-60) “that bees provide to wildflowers and crops”

Comment: L56 , but

Reply: L59. We replaced ‘but’ with ‘and thereby’ and changed other parts of the last sentence of the abstract (see the point immediately below).

Comment: L56: So full disclosure I’ve never really bought into this urban pollination argument as it seems a bit contrived to me. That aside, when you talk about pollination services its got to be related to something, crops are an obvious one, but wild plants are clearly another (and I would go out of my way to argue here that it needs to be native rather than non native plants which ultimately represent potentially invasive species). So to justify the ‘increases pollination services line’ I think it needs to be clear here who benefits. If there are either 1) lots of crops in cities (which in general their aren’t) or 2) lots of native plant species, ideally rare ones (again I am not clear that there are, but there may be) this would be a justification of why cities are so great. I would like this clarification, justification or caveat made clear somewhere if you haven’t already don’t this, and while it would be nice in the abstract I do see space as a problem.

Reply: We have two responses to this comment. Firstly, German cities are considered hotspots with regard to native plant species (Kuhn et al. 2004, Knapp et al. 2008) and their pollination is therefore of relevance for nature conservation. Secondly, there is increasing interest in the potential of urban agriculture (Lawson, 2016) (two links to current initiatives in German cities:

<https://foodtank.com/news/2014/03/ten-urban-agriculture-projects-in-berlin-germany/>; <https://germanysustainablecommunities.wordpress.com/urban-agriculture/>). Thus crop pollination is also an important component of the ‘ecosystem service of pollination’ in urban areas, about which we write. In the abstract and in the main manuscript, we refer to the pollination service that bees provide to both wildflowers and crop plants in cities. We have now revised our abstract. L57-L60. “Appropriately managed cities could enhance the conservation of Hymenoptera and thereby act as hotspots for pollination services that bees provide to wildflowers and crops grown in urban settings”

Kühn, I., Brandl, R. & Klotz, S. The flora of German cities is naturally species rich. *Evol. Ecol. Res.* 6, 749–764 (2004).

Knapp, S., Kühn, I., Schweiger, O. & Klotz, S. Challenging urban species diversity: contrasting phylogenetic patterns across plant functional groups in Germany. *Ecol. Lett.* 11, 1054–1064 (2008).

Lawson, L. Agriculture: Sowing the city. *Nature* 540, 522–523 (2016)

Later in our introduction we discuss the importance of pollination service provision not only in natural and agricultural but also in urban ecosystems. We furthermore discuss the importance of pollination for both native flowering plants and crops in urban areas. L80-L90. “Pollination is a crucial ecosystem service not only in natural but also in agricultural and urban ecosystems. An estimated 87.5% of angiosperm species are dependent on animal vectors for pollination and seed set¹³ whilst the

current economic value of pollination to world agriculture is ca. US\$ 235-557 x 10⁹ per annum at 2015 prices¹⁴. Diverse land uses within European cities can be very rich in native flowering plant species^{15,16} and there is also an increasing interest in the potential of (outdoor) urban agriculture in ensuring food security¹⁷. Yet the impact of urbanization on the pollination of wild and cultivated plants remains poorly known¹⁸. We also lack direct comparison with rural sites, which are typically dominated by agricultural land use and where pollinators are vital for crop pollination, yet where agricultural intensification is thought to result in reduced provision of a range of ecosystem services provided by insects, including pollination¹⁹.”

Introduction

Comment: L70: Ok, so the biodiversity thing is one thing, but I would really like to see some clear justification of the pollination ES component in relation to cities. See points above. Note simply saying the may spill over to me would need some convincing given their foraging ranges would mean that from a large conurbation this would only be say a couple of km around the edges – not necessarily relevant for the countryside in general (say in comparison to a spread of semi-nat habitat though the countryside). I am sure you can put an argument together. Ultimately I take your point on the L72 that a lot of agricultural land is shit for pollinators, and that there is sig potential for urban landscape to support diversity, but ultimately bees in agriculture landscape are providing crop pollination services. Again if you can show a prevalence of native species of pant in cities then the pollination element here is more justifiable.

Reply: We agree that, at any one point in time, a bee pollinator nesting in a city will only visit flowers (i.e. pollinate) in its immediate vicinity and therefore cannot itself contribute much to large-scale agricultural field-crop pollination (production), unless fields are immediately adjacent to the city. But by acting as sites where bees can breed, cities can act as sources of new individuals in the next generation (offspring) that can disperse to agricultural areas (even if such areas are potential sinks for bees), where they can pollinate. For hoverflies, many of which disperse over long distances, cities really could act as an important source of fly pollinators for distant agricultural areas within the same season. But a more compelling argument of immediate relevance is that cities require pollinators because they retain many wild plant species and are of growing importance for (urban) food production, as we have specified in our response to the point immediately above this one (where we also cite relevant text from the revised ms).

Comment: L80 – Ok so expand on this – if you are talking about city gardens like they have in cuba, I get it. If its vertical faming that’s more insulated form wild pollination (more likely to use greenhouse pollinator systems like B. terrestris colonies).

Reply: We indeed refer to outdoor urban farms and gardens that are used in many European cities for local food production. We have now revised our manuscript.

L83-L86 *“Diverse land uses within European cities can be very rich in native flowering plant species^{15,16} and there is also an increasing interest in the potential of (outdoor) urban agriculture in ensuring food security¹⁷.”*

Comment: L120; Isn't it used more though as a justification for why we focus on phylogenetically distinct species like gorillas as (where losing a species is significant) compared to say butterflies (where we lose species every day and no one really cares. To be fair, it's a matter of scale I suspect.

Reply: We concur that, in communities where species have high functional trait overlap due to recent shared evolutionary history, each individual species contributes less to overall community function (the corollary: species with disparate evolutionary history will cover broader trait space). OK, gorillas, pandas and elephants are phylogenetically distinct and therefore all deserve conservation attention. But even the bees, for example, encapsulate over 120 million years of evolutionary history and display a wide range of 'functional' traits such as tongue length and the ability to open poricidal anthers. We therefore think it is valid to examine phylogenetic distinctiveness of bees and their communities, an approach which might also prove valuable in conservation assessments. We've tried to address this point by examining phylogenetic diversity using two metrics, which quantify the evolutionary history represented within a community, which capture similarities in traits that mediate responses to the environment and which potentially reflect similarities among taxa in the traits that contribute to ecosystem function. By quantifying phylogenetic diversity in our study, we were able to address its importance as a predictor for the ecosystem service of pollination – which it was – as has recently been documented elsewhere (Grab et al. 2019). Our data therefore support the view that broad taxonomic coverage (and presumably broad 'functional' trait coverage) is related to improved pollination.

Grab, H. et al. Agriculturally dominated landscapes reduce bee phylogenetic diversity and pollination services. *Science* 363, 282–284 (2019).

Comment: L132-139: some of this detail seems more suited to the methods (possibly you were asked by another reviewer to put it in there).

Reply: We agree with the reviewer and we have now shortened the paragraph to give the essentials of our study without justification for the methods employed, all of which are given in the Methods section.

Methods

Comment: L148: Do you really need to say 'statistically powerful' – its just a blocked design, your not reinventing the wheel or anything.

Reply: True; our hubris. We have now revised our methods. L160-L162. "To test the association between urban versus rural land use on insect pollinators and pollination, we used a well-replicated study design that employed a flower-rich urban site paired"

Comment: L159: I could do with some more description of these rural sites, as at the end of the day the comparison lives or dies on the basis of the sites you chose. I think giving some more clear criteria (even if the detail is in an appendix) would be useful here. Ultimately a different set of criteria for the rural sites could have yielded very different results. I don't have an issue with your design, but that caveat would be good. It also crosses my mind you may want to justify the benefits of your approach over or pros/cons of the obvious alternative design of an urbanisation gradient.

Reply: We agree that selection of sites is critical to the interpretation of our results. We aimed to sample at optimal (for insect pollinators) city sites and optimal (for insect pollinators) rural sites, thereby comparing the best with the best. Based on former studies (i.e. Baldock et al. 2019, Theodorou et al. 2017) and our own experience, we selected the most promising city sites (botanical gardens, amenity parks) that (i) contained a lot of flowers and (ii) were not covered in hard surfaces (though they are often the subject of intense management, weeding, digging over etc.). We then paired each city site with an equivalent and most promising flower rich rural site, namely with seemingly permanent semi-natural vegetation. We now make more explicit our logic and criteria for site selection and extend information on the criteria we used to select our rural sampling sites. We also added to our text (Methods) L174-L177. *“Our logic was to compare the best urban sites for insect pollinators^{16,22} with the best rural sites that were matched in terms of habitat structure (land cover, flower abundance), though each pair was sited in a different land use matrix (urban vs. rural, respectively).”*

Baldock, K. C. R. et al. A systems approach reveals urban pollinator hotspots and conservation opportunities. *Nat. Ecol. Evol.* **3**, 363–373 (2019).

Theodorou, P. et al. The structure of flower visitor networks in relation to pollination across an agricultural to urban gradient. *Funct. Ecol.* **31**, 838–847 (2017).

For rural sites, details read L193-L209. *“Rural sites were selected to be as similar as urban sites in terms of their local (250 m scale) land cover (i.e. flower abundance) by using land cover maps within a Geographic Information System (Quantum GIS). Arable land (=agricultural land) and forest/semi-natural cover were the dominant land use types at the landscape scale, comprising a mean of 45% and 41% the of surrounding (1,000 m scale) land use across all rural sites, respectively (Supplementary Table 3), typical of the region's rural environment i.e. our rural sites were not impacted by urban sprawl. To identify rural sites that were ideal for insect pollinators yet independent of urban sites, we drew a buffer at a circumference of 10 km radius from each urban site and then used GIS to identify areas of semi-natural grassland and scrub vegetation immediately outside the buffer that were largely devoid of 'residential' and 'commercial/industrial' and dominated by arable land and forest/semi-natural cover within the surrounding 1 km radius. Candidate rural sites were then visited and one was randomly selected that had a 25 × 25 m area near its geographical centre with diverse floral resources (Supplementary Table 1), which we used as our rural sampling plot. By using these criteria for site selection, we aimed to*

sample from the best sites for insect pollinators, and potentially also for pollination, among urban and among rural localities.”

We used a paired design in order to address the main question of our study: are there any differences in insect pollinator richness and pollination service provision between flowering plant rich urban vs. rural areas? The study of insect biodiversity and pollination along a gradient (or transect) of urbanisation (e.g. in regards to impervious surfaces etc.) would have required a 10 fold increase in sampling effort and would have addressed a slightly different question, namely what are the effects across a gradient of urbanisation on insect communities and the pollination they provide? We note that there have been a number of articles addressing aspects of just this question e.g. Bates et al. 2011, Fortel et al. 2014, Theodorou et al. 2017. We now add to the manuscript at L178-L182. “Though sampling across a gradient of urbanisation and into the rural landscape is another sampling strategy that has been used to demonstrate the importance of cities for pollinators^{20–22}, our intention was to compare urban with rural habitats and thus we maximised the number of urban-rural comparisons for our given intensity of sampling.”

Bates, A. J. et al. Changing bee and hoverfly pollinator assemblages along an urban-rural gradient. *PLoS One* 6, e23459 (2011)

Fortel, L. et al. Decreasing abundance, increasing diversity and changing structure of the wild bee community (Hymenoptera: Anthophila) along an urbanization gradient. *PLoS One* 9, e104679 (2014).

Theodorou, P. et al. The structure of flower visitor networks in relation to pollination across an agricultural to urban gradient. *Funct. Ecol.* 31, 838–847 (2017).

Comment: L166-168: So maybe you talk about this later, but if not I would like you to do this, however it strikes me you have a honeypot situations going on here. Do you have a high (phylogenetic) diversity, or is at a sampling effect of what bees there are being concentrated into a small patch of limited resources. It the difference between quantifying diversity in a 2 x 2m quadrat in a field of oilseed rape vs. the same area of oilseed rape on its own in an area otherwise relatively impoverished of flowering plants (e.g. a 2x 2 m area of OSR in a wheat field). It’s likely you may see only a handful of bees in the quadrat in the OSR field, but many more species in the isolated patch as they are focusing in on this resources from a large area. Again, doesn’t negate what you have done, but it seems a potentially confounding factor.

Reply: This is a good point that we have not explicitly addressed. We agree with the reviewer that, indeed, concentration/honeypot (nice term) (or potentially the opposite – dilution effects?) might influence biodiversity estimates we made at our sites in either of the ecosystems, rural and urban. But we argue that the honeypot effect, if it exists, is likely to be as true in the rural sites as it in the urban sites.

Firstly, we used sites that were at least 25x25 m with diverse floral resources. This ensured at least a minimum patch size across our sampling sites (urban and rural), and floral abundance did not differ between ecosystems (though floral diversity

did slightly), so the ‘pot’ of the ‘honeypot’ was similarly large (or similarly small) in both urban and rural sites. Secondly, the landscapes surrounding urban and rural sites did not differ markedly in potentially insect-relevant landscape features such as green cover and edge density (Supplementary Figure 4), both of which might be considered suitable for bee pollinators (indeed, edge density was the single landscape factor that in our SEMs was associated with both Hymenopteran diversity and pollination in both urban and rural sites). Our verbal argument is thus that rural sites were in a matrix of forest and cropland which, during our sampling period (mid-summer), had little to no flowers, much like roads or concrete paving at urban sites; both were likely equally impacted by the honeypot effect. We have now raised the potential for a honeypot effect in the Methods at L404-L409.

“Flower-rich sites located within an inhospitable landscape may attract insects from further afield than sites nested within a floristically rich landscape, a ‘honeypot’ effect. Though we did not quantify floral abundance across the wider landscape so as to test explicitly for the honeypot effect, we tested for differences between urban versus rural sites in terms of local (flower abundance, flower diversity) and landscape variables (green cover, edge density) using LMMs and GLMMs to assess whether they varied consistently between ecosystems.”

We have also mentioned in our Results the honeypot effect likely did not play a role at L518-L530 (for insect communities). *“The honeypot effect might in part account for differences we detected between urban versus paired rural sites in insect pollinator community diversity. Yet local and landscape covariates included in our statistical models did not differ markedly between site type. Firstly, total flower abundance in a 200 m buffer around each 25 x 25 m site did not differ between urban versus rural sites (LMM, $t=0.403$, $P=0.697$; Supplementary Table 1). Even though species richness of flowering plants was higher at urban sites (GLMM, $Z=3.350$, $P<0.001$; Supplementary Table 1), our data suggest that urban and rural sites were similar in their capacity to attract flying insects from afar. Secondly, landscape factors that might be particularly associated with high insect community biodiversity, namely total green cover and edge density (see below), did not differ between urban versus rural sites (LMM, $t=-0.080$, $P=0.938$; LMM, $t=0.487$, $P=0.632$, respectively; Supplementary Fig. 4). These results suggest that, if a honeypot effect had impacted the insect communities that we measured, then it likely impacted both urban and rural sites equally.”*

For pollination, we added (L577-L582) *“We cannot exclude a honeypot effect having led to greater *T. pratense* pollination in urban versus rural sites. However, as described in our analysis of insect community composition above, surrounding land use of urban and rural sites was equally inhospitable for flying insect pollinators (Supplementary Fig. 4). The honeypot effect is likely to have operated at both urban and rural sites.”*

But we acknowledge that we have not explicitly tested for a honeypot effect, which is probably difficult to do retrospectively and which would likely be best addressed by using a landscape scale experiment in which flowers are selectively grown, e.g. wildflower strips, or selectively cut across wide swathes of the landscape. We have therefore added a further caveat to the Discussion L672-L677

“Though we argue that the honeypot effect did not impact our study’s response variables (insect biodiversity and pollination) because local and landscape-level ecological variables related to flying insect pollinators did not vary markedly between urban versus rural sites, we cannot formally exclude it. Replicate, landscape-level experiments selectively increasing or decreasing flower abundance and diversity might offer one option to test for its effect on insect diversity and pollination.”

Comment: L171: so forest / semi-nat covers a pretty broad brush of habitats. Within this spectrum I would have thought say species rich grassland dominated would be better for bees than forest dominated. Was that an issue?

Reply: We agree with the reviewer that European forests do not offer resource rich habitats for bees (but they might do so for other insects e.g. beetles). We only used the category forest/semi-natural in first demarcating localities > 10 km from our urban sites that could potentially be used as rural sites. We then indeed selected rural sites based on the presence of semi-natural flower-rich (=grassland/shrub) vegetation, which we now clarify. L199-L207 *“To identify rural sites that were ideal for insect pollinators yet independent of urban sites, we drew a buffer at a circumference of 10 km radius from each urban site and then used GIS to identify areas of semi-natural grassland and scrub vegetation immediately outside the buffer that were largely devoid of ‘residential’ and ‘commercial/industrial’ and dominated by arable land and forest/semi-natural cover within the surrounding 1 km radius. Candidate rural sites were then visited and one was randomly selected that had a 25 × 25 m area near its geographical centre with diverse floral resources (Supplementary Table 1), which we used as our rural sampling plot.”*

Then, in our analyses, we separately quantify forest cover (i.e. coniferous, deciduous and mixed forest) and semi-natural cover (i.e. grassland, meadows and shrubland) and several other landscape scale metrics as potential predictors when investigating the main drivers of insect richness within each ecosystem as well as when comparing ecosystems. L357-L365. “Several metrics known to impact flying insect biodiversity were used to quantify landscape heterogeneity (composition and configuration) at each of the 18 sites at both 250 m and 1,000 m scales^{31,32}. These were (i) the proportion of semi-natural cover (grassland, meadows and scrub vegetation), (ii) the proportion of forest, (iii) the extent of arable (=agricultural) cover, (iv) the proportion of residential and (v) commercial/industrial land cover, (vi) the extent of botanical gardens, public parks and allotments, and (vii) edge density, as total length of ‘green cover’ (semi-natural and forest cover, botanical gardens, public parks, and allotments) patch edges divided by the total area, and which represents a quantification of landscape configuration.”

Neither forest nor semi-natural cover were important predictors of insect richness. Habitat diversity (Shannon diversity of land-uses), local flowering plant richness, edge density and proportion of arable land were the best predictors of insect richness in rural ecosystems (see figure 3).

Comment: L199-202: 5 consecutive days of sampling – all in one - seem problematic to me. What about temporal variability across the year. I would have preferred say 3 days of sampling, but separated by a month each. Do you think this will impact the results?

Reply: The main aim of our study was to compare (communities of insects and pollination) between rural and urban ecosystems. We achieved this by using a paired design and a common methodology of sampling and experiments across ecosystems. Increasing our sampling effort per site across the season would have indeed provided more information on the insect fauna in each site but we think this would not have impacted our results. With our 5 consecutive days of sampling at any one pair of sites, and across 9 pairs of sites, we could (i) ensure saturation of sampling at any one pair of sites and (ii) also have enough power to detect consistent differences in species richness between ecosystems, without having to take into account the potentially confounding issue of insect phenology. We nevertheless appreciate the point that sampling across a longer time period (preferably across the entire year) would give an even more accurate estimate of insect community diversity, and we mention this in the Discussion (L779-L781). *“As we collected insects for only 5 days per site, intensifying sampling across the year would also ensure that the urban/rural effect we detected in our data is robust across the phenology of temperate insects.”*

Comment: L237: give a reference for the link between phyl and trait diversity. If I am honest if you are explicitly just measuring phyl diversity I would just refer to this, and maybe in the discussion mention the two are correlated. If you are explicitly deriving trait diversity fair enough.

Reply: This is a fair point because we indeed only measured phylogenetic diversity and inferred trait diversity. We now provide a reference (see below) on the link between phylogenetic and trait diversity: Tucker *et al.* (2018).

Tucker, C. M., Davies, T. J., Cadotte, M. W. & Pearse, W. D. On the relationship between phylogenetic diversity and trait diversity. *Ecology* 99, 1473–1479 (2018).

Comment: L254: So I have already expressed my feeling that you need to justify the Es approach more. It's also clear that you have picked a flower that has a structure that precludes shorter tongued species. I accept you can't do everything, but again this should be made clear that this is not necessarily the best phytometer for many species, who may be excluded from using it. Flies jump to mind here as being obvious ones, but I would have thought that shorter tongued solitaries would also not really pollinate this. I get you justification with the correlations, which goes somewhat to justify this (L258) but presumably this just means there were lots of *Bombus* in all communities rather than it's a good model for all species.

Reply: In our methods L.273-L280 we discuss why we used red clover as a pollinometer to quantify pollination service provision in rural and urban ecosystems. But we understand the concerns of the reviewer because we only infer that high red

clover pollination equates with high overall pollination (including by short-tongued insects) at our 9 paired sites, which is based upon our earlier study (Theodorou et al. 2017). We have therefore introduced an additional caveat to our text. L750-L762 (in the Discussion): *“Recent studies support the view that pollination service provision is enhanced by high pollinator species diversity⁷⁴, including high pollinator phylogenetic diversity⁴¹. We nevertheless urge caution in the interpretation of our results because flowers of our pollinometer *T. pratense* plants have long corollae and were mainly visited by (long-tongued) bumble bees. Thus the differences between urban and rural sites in pollination that we recorded were likely causally related to *Bombus* visitation rates, which were higher at urban sites. Though we found Hymenoptera OTU richness to be higher in urban compared to rural areas, as also seen by other studies^{23,57}, and though in a previous study we found a high correlation between pollination service provision to *T. pratense* and to three other plant species, including those with open flowers²², a more comprehensive set of pollinometer species covering diverse floral morphologies is needed to test unequivocally the role of pollinator species richness versus pollinator phylogenetic diversity in enhancing community-wide pollination.”*

Incidentally, we use the term pollinometer rather than phytometer, which was a suggestion from the editor of our PRSL-B paper (Theodorou et al. 2016), Steve Johnson. Because we measure pollination in experimental plants and not another aspect of the plant e.g. photosynthesis, pollinometer seems appropriate. But if the referee insists, we would be happy to revert to phytometer to describe our potted red clover plants as we understand that the term is widely used in the literature to describe experimental plants located in the wider environment.

Comment: L275: So my point above about the honeypot issue in cities (i.e. an isolated patch of flowers attracting in lots of bees (including honeybees which are often kept in cities and will really narrow in on these areas) to a small patch in a sea of concrete compared to the diluting effect of a rural landscape with lots of flowers is also very relevant here for the phytometers. How do you deal with this problem – I would have thought it really causes potentially large problems with this methods. If you limited it to virgin flowers and say X visitations by pollinators before bagging, at least this would be standardised. That said that sounds horrific to do, so maybe just caveat the issue here so it’s clear this is trends may be a potential artefact.

Reply: This is a good point, though it is not clear whether the honeypot effect will be any greater (or lesser) in urban versus rural sites. As we have commented above, local flowering abundance, availability of red clover pollen donors and potential land-uses that might have affected our biodiversity and seed set estimates were quantified and used as covariates in analysis of differences in biodiversity and pollination between urban and rural sites, potentially accounting for a honeypot effect. Moreover, we used potted experimental plants, bagged virgin flowers and estimated the confounding effects of visitation rates when we investigated the effects of phylogenetic diversity (PSV, MNTD) on pollination service provision. Such analyses could have indicated whether in our areas we had any dilution or concentration effects, which apparently we did not (‘drivers’ of biodiversity and of pollination were similar in both habitat types, as seen in SEM

results for urban, rural, and combined datasets). We have nevertheless acknowledged that a honeypot effect cannot be excluded (from either urban or rural sites) and therefore have phrased a caveat to this effect in the Discussion. L672-L677. “Though we argue that the honeypot effect did not impact our study’s response variables (insect biodiversity and pollination) because local and landscape-level ecological variables related to flying insect pollinators did not vary markedly between urban versus rural sites, we cannot formally exclude it. Replicate, landscape-level experiments selectively increasing or decreasing flower abundance and diversity might offer one option to test for its effect on insect diversity and pollination.”

Comment: L290 – Ok good, in relation to my above comment you have considered this -I would still explain the potential risks linked to this though.

Reply: Assuming this point relates to the honey pot effect, we have addressed it above and provided additional comment and a caveat to our Methods, Results and Discussion sections.

Comment: L320-327: Fair enough, it’s a reasonable approach to get a sensible scale for you landscapes.

Reply: Thank you.

Comment: L421: So my understanding (and correct me if I am wrong) was that structural equation models required huge numbers of replicates to produce anything reliable, you don’t actually seem to have that in your study. I am guessing there are metrics for describing the reliability of model outputs, it would be good to explain how you are using them here. Is this what L439 is doing?

Reply: We agree with the reviewer that SEM is a data hungry method. Due to this, we used multiple linear mixed models to build the SEM hypothesised paths. We then used a backwards stepwise elimination process based on AICc to remove non-significant pathways. The available literature on the topic suggests this to be the most robust and justifiable approach to SEM. The fit of our final model was then assessed using a Fisher’s C statistic, suggesting stable fit to our data. We therefore have confidence that our SEM results are robust.

Results

Comment: Happy with these

Reply: Thank you.

Discussion

Comment: So I am happy with the discussion – what I want is the limitations of what you have done to be made clear. I can see this has undergone extensive review and while I don’t want you to start from scratch I do see some of the approaches as

having the potential to create artefacts. As long as these limitations are clear this works for me.

Reply: In our above responses (and additions to the ms), we have addressed these major concerns: (i) who benefits from ES in cities, (ii) limitation of the *Trifolium* pollinometer, (iii) choice of rural sites and (iv) honeypot issue.

Comment: L564: this may be overplaying it, its 9 cities in Germany. Its undeniably a C.European landscape but this would suggest a cross country scope of the study sites to me.

Comment: This is true, and we have now included in the manuscript potential limitations of our study.

In the abstract, we have qualified the scope of the study by adding (L50) "Central" before "European cities"

Furthermore, we have revised Discussion L646, which now reads: "In this replicated study across the Central European landscape, we found that the..... "

Reply: We thank the reviewer for their many insightful comments.

Reviewer #5 (Remarks to the Author):

I am primarily providing a technical evaluation of the metabarcoding methods in the revised manuscript by Theodorou et al "Flower-rich urban areas can act as hotspots for bees and the ecosystem service of pollination but are not a panacea for all insects"

METABARCODING METHODS (BASED ON VEIWING OF THE REVISED MANUSCRIPT ONLY)

Comment: The metabarcoding approach used is a bit dated (large PCR fragments, 454 technology), but methods are done to a standard that has been common in the literature. The use of only 18 samples (one per site Line 122 supporting material) is unfortunate. Insects were sampled on 5 consecutive days and each day could have been analysed separately (90 samples) to give extra insight.

Reply: We thank the reviewer for acknowledging our robust meta-barcoding pipeline. Insect OTUs richness accumulation curves indicate saturation in each of our sites, suggesting that our sampling and metabarcoding methods captured most of the community diversity (see Supplementary figure 6, given below the following point). Thus analyzing separately each of our sampling day material most likely would have not provided extra information.

There are also few pieces of data useful for interpretation that are missing:

Comment: - The final number of sequence reads per sample and rarefaction curves should be reported in the supporting material. The reported lack of correlation between read number and diversity (line 234) seems to be hand waving.

Reply: We now provide the total number of reads per sample (Supplementary Table 12) and rarefaction curves (Supplementary Figure 6). Insect OTU richness accumulation curves indicate saturation in each of our sites, suggesting that our sampling and metabarcoding methods captured most of the community diversity. In addition we correlated OTU richness with rarefied and extrapolated (Chao1) total OTU richness. All these metrics were highly correlated ($P < 0.001$; Supplementary Table 13).

Supplementary Figure 6. OTU accumulation curves per site.

Supplementary Table 12. Number of reads and OTU richness per site.

Site	Number of reads	OTU richness
Rural Halle	6475	76
Urban Halle	6732	80
Rural Leipzig	4139	122
Urban Leipzig	5282	93
Rural Jena	3877	105
Urban Jena	3417	68
Rural Dresden	4344	90
Urban Dresden	3528	53
Rural Chemnitz	6729	77

Urban Chemnitz	4569	57
Rural Braunschweig	4392	81
Urban Braunschweig	5133	73
Rural Berlin	3561	74
Urban Berlin	5080	61
Rural Potsdam	4670	71
Urban Potsdam	4183	71
Rural Göttingen	4723	83
Urban Göttingen	4077	62
Total	8 4911	

Supplementary Table 13. Pearson correlation coefficients (r , below diagonal) of the relationship between detected OTU richness, rarefied OTU richness and extrapolated total OTU richness (Chao 1), and significance (uncorrected P values, above diagonal).

	Detected OTU richness	Rarefied OTU richness	Chao1
Detected OTU richness	1	<0.001	<0.001
Rarefied OTU richness	0.9802	1	<0.001
Chao1	0.9830	0.949	1

Comment: - The bioinformatics pipeline was evaluated with data from another exemplar study, and for what it's worth the results appear robust. For completeness it is also important to report the number of OTUs recovered that didn't match the reference Sanger OTUs and number of recovered OTUs that matched the same reference species.

Reply: We now provide the number of OTUs recovered that didn't match the reference Sanger OTUs in Supplementary Table 14 (also given below). Overall the numbers are very low, suggesting that our pipeline is very robust.

Supplementary Table 14. Number of Sanger sequence-generated OTUs (Sanger OTUs) for each mock community and 454-generated OTUs successfully blasted to Sanger OTUs at 97% similarity, using both the original pipeline of Yu et al.¹⁰ and our pipeline, as well as OTUs we detected with our pipeline that did not match those of Yu et al.¹⁰.

Mock communities	Sanger OTUs	≥1-read OTUs Yu et al. 2012	≥1-read OTUs this study	OTUs that did not match the reference
1H1X	159	107 (67.3%)	129 (81.1%)	13
XSBN	230	156 (67.8%)	168 (73.0%)	7
KMG	152	127 (83.5%)	133 (87.5%)	11
HongHe	167	133 (79.6%)	147 (88.0%)	12
2H1K	140	117 (83.5%)	129 (92.1%)	13
2K1X	134	90 (67.1%)	103 (76.8%)	11
5K1X	106	67 (63.2%)	75 (70.7%)	5
All communities	547	408 (74.5%)	484 (88.5%)	

Comment: Regardless of potential technical issues, the same metabarcoding methods were being applied across the sites, so even if there are some biases, the comparisons of alpha diversity should be valid.

Reply: We agree with the reviewer.

Comment: While the metabarcoding approach is good, the study would have been considerably stronger in my mind if supported by some morphological identification and some measure of insect group relative abundance. The real strength of metabarcoding applications is in systems that are very diverse, or where identification isn't feasible – this doesn't seem to be true in this study.

Reply: We agree with the reviewer that combining the use of conventional morpho-taxonomic and metabarcoding approaches will be, of course, the optimum. Nevertheless, morphological identification of large trap catches of mixed insect communities like ours will have required the involvement of people with a high level of taxonomic knowledge of specific taxonomic groups to achieve low human error rate. The German barcode of life initiative to barcode all animals runs into difficulty with the true flies because there is no taxonomic expert of many Dipteran groups, resulting in many 'unknown' Diptera in other metabarcoding analyses (e.g. Geiger et al. 2016). Our study aim was to compare the two ecosystems (rural vs urban) in regards to overall biodiversity across multiple insect orders using a standardised sampling and analytical method. We achieved that with the use of metabarcoding and would not have gotten as close in quality to our dataset using traditional morphological based taxonomy, especially with the Diptera.

Geiger, M. *et al.* Testing the Global Malaise Trap Program – How well does the current barcode reference library identify flying insects in Germany? *Biodivers. Data J.* (2016). doi:10.3897/BDJ.4.e10671

Comment: An additional analysis using the current dataset that would strengthen the manuscript is inclusion of some measures of beta-diversity. Are the urban communities more similar to each other, or to the communities in the neighbouring rural sites? Are the same hymenopteran species being boosted in all the urban centres?

Reply: This is a good idea for how to explore our metabarcoding data more fully. We have now undertaken and added the additional analyses of community composition in the manuscript, both to our methods and our results. **Methods: L424-L435** *"To test for differences in insect community composition between flower-rich urban and rural sites, we performed a paired permutational multivariate analysis of variance using the adonis function, with 999 permutations, implemented in the R package vegan* ⁶¹. *In the adonis analysis, the Jaccard distance matrix of species composition was the response variable, with ecosystem (urban/rural) as a fixed factor. The strata (block) argument was set to 'site pair' so that randomizations were*

constrained to occur within each pair and not across all sample sites. We undertook these analyses for all insects and for Hymenoptera only. We employed non-metric multidimensional scaling (NMDS) within the package *vegan* to visualize the variation in community composition. For each site we also calculated the mean ecological distance (Jaccard index) over all pairwise comparisons of the 9 sites belonging to the same ecosystem type. We used a LMM to compare urban and rural ecosystems, with pair as a random effect factor and using all local patch and landscape variables as covariates.”

Results L556-L564. “Both total insect composition and Hymenoptera community composition differed between rural and urban ecosystems (adonis all insects: $F=1.574$, $R^2=0.089$, $P=0.003$; Hymenoptera only: $F=1.692$, $R^2=0.095$, $P=0.004$, respectively; Supplementary Fig. 5); several species were found in both urban and rural sites e.g. *Bombus terrestris* and *Lasioglossum calceatum*, but others were restricted to few sites, often within the same ecosystem e.g. *Heriades truncorum* in urban sites, *Dasygaster hirtipes* in rural sites (see Supplementary Dataset). Overall insect communities and Hymenoptera communities were more homogeneous in urban compared to rural sites (LMM, all insects: $t=2.587$, $P=0.032$; only Hymenoptera: $t=4.312$, $P=0.002$; Supplementary Fig. 5).”

Supplementary Figure 5. Non-metric multidimensional scaling (NMDS) ordination of (a) overall insect communities and (b) Hymenoptera communities. Stress levels are reported in the top right of the figure. Results of the *adonis* analyses of differences in community composition are reported in the top left of the figure.

Comment: The experiment represents a considerable amount of work, careful analysis and presentation. The relatively limited geographical scale of the study has mixed impact. It may have strengthened the conclusions – since the urban and rural landscapes being compared are likely to have a similar signal. However, this does necessarily constrain the generality of the findings (i.e. do the findings hold outside of Europe).

Reply: We agree with the reviewer that our study addresses Central European

(German) cities and surrounding rural landscapes and therefore we should be cautious in extrapolating our results. We have therefore added caveats to this effect to the abstract: L50 “Central” added before “European cities”

In the Discussion, we also commence with. L646. “In this replicated study across the Central European landscape, we found that the..... ”

Nonetheless, we feel that our results make even more pressing and timely the main messages from our study, which ought to be considered and incorporated by the architects of city environments, new (in Asia and Africa) and old (in Europe).

COMMENTS ON RESPONSES TO PREVIOUS REVIEWER COMMENTS

Comment: Two of the weaknesses I raised above (the value morphological identification would have added; limitation on conclusions due to the relatively small geographic scale) were raised previously, but the authors can't make changes to improve these aspects of the study.

Reply: Indeed we agree with the reviewer regarding the value of morphological identification; we understand that researchers may prefer traditional morphotaxonomy based bio-assessment, or a mix of the two. However, we argue that metabarcoding has been shown to be faster, cheaper and more comprehensive compared to traditional morphological taxonomy^{10,12-14}. Throughout our ms, we have been careful in clarifying that our study addresses German cities (and surrounding rural landscapes), and we have added a caveat about the scale of our study.

Comment: The technical responses to Reviewer 2 on metabarcoding seem reasonable. As outlined above the methods used in the study are slightly dated, but the analysis performed here are defensible. Several aspects of data analysis were improved in response to the reviewers' comments. I agree with the authors' sentiment that there are ongoing debates regarding bioinformatics and OTU clustering, and this fine-tuning will be unlikely to affect their comparisons of species diversity. Looking at iNEXT for rarefaction curves would be useful even if the authors don't extrapolate species richness.

Reply: We thank the referee for the comments. We agree with the reviewer that plotting rarefaction curves using *iNEXT* or any other R package i.e. *vegan* provides a valuable visual representation of the level of saturation of OTUs richness and number of reads. We now provide rarefaction curves in the supplementary material (Supplementary Figure 6). The insect richness was saturated in each library, suggesting that our sampling and metabarcoding method captured most of the community diversity. In addition we correlated OTUs richness with rarefied and extrapolated (Chao1) total OTUs richness. All these metrics were highly correlated ($P < 0.001$; Supplementary Table 13).

Comment: The metabarcoding component of the paper is suitable for publication, but it does not strengthen the paper. Comments on the significance of the main conclusions are best left to other reviewers.

Reply: We thank the reviewer for their many helpful comments.

Reviewers' Comments:

Reviewer #4:

Remarks to the Author:

Firstly I thought the original manuscript was well written, if a little long. A lot of my comments focused on the need for clarification of a number of points - in particular a direct justification of the merit of urban systems (at least as a response to me). So having gone back through my original comments to the author I can see that I went a bit overboard on this. That said I am more than happy with the reviewer responses, either where they have changed or clarified the manuscript or justified where you felt my point of view was incorrect. While I may not agree with everything, that's not really the point of this, rather you have provided a solid and interesting manuscript that in my opinion is of high enough quality and novelty to merit publication in this journal. I think this will make a valuable contribution to the literature and hope you don't take too personally my original review. A great job.
Ben Woodcock

Reviewers' Comments:

Reviewer #4 (Remarks to the Author):

Comment: Firstly I thought the original manuscript was well written, if a little long. A lot of my comments focused on the need for clarification of a number of points - in particular a direct justification of the merit of urban systems (at least as a response to me). So having gone back through my original comments to the author I can see that I went to town a bit on this. That said I am more than happy with the reviewer responses, either where they have changed or clarified the manuscript or justified where you felt my point of view was incorrect. While I may not agree with everything, that's not really the point of this, rather you have provided a solid and interesting manuscript that in my opinion is of high enough quality and novelty to merit publication in this journal. I think this will make a valuable contribution to the literature and hope you don't take too personally my original review. A great Job. Ben Woodcock

Reply: We thank the referee for their positive comments.